# Cashew (*Anacardium occidentale*) Extract: Possible Effects on Hypothalamic–Pituitary–Adrenal (HPA) Axis in Modulating Chronic Stress

**DOI:** 10.3390/brainsci13111561

**Published:** 2023-11-07

**Authors:** Guedang Nyayi Simon Désiré, Foyet Harquin Simplice, Camdi Woumitna Guillaume, Fatima Zahra Kamal, Bouvourné Parfait, Tchinda Defo Serge Hermann, Ngatanko Abaissou Hervé Hervé, Keugong Wado Eglantine, Damo Kamda Jorelle Linda, Rebe Nhouma Roland, Kamleu Nkwingwa Balbine, Kenko Djoumessi Lea Blondelle, Alin Ciobica, Laura Romila

**Affiliations:** 1Department of Biological Sciences, Faculty of Science, University of Maroua, Maroua P.O. Box 814, Cameroonharquin.foyet@univ-maroua.cm (F.H.S.); gcamdi77@gmail.com (C.W.G.); parfaitoel@gmail.com (B.P.); shtchinda@gmail.com (T.D.S.H.); eglantinewado@gmail.com (K.W.E.); kenkolea@gmail.com (K.D.L.B.); 2Laboratory of Physical Chemistry of Processes and Materials, Faculty of Sciences and Techniques, Hassan First University, B.P. 539, Settat P.O. Box 26000, Morocco; 3Preclinical Department, Apollonia University, Păcurari Street 11, 700511 Iași, Romania; 4Center of Biomedical Research of the Romanian Academy, 700506 Iasi, Romania; 5Department of Biology, Faculty of Biology, Alexandru Ioan Cuza University, 11 Carol I Blvd., 700505 Iasi, Romania

**Keywords:** *Anacardium occidentale*, depression, stress, monoamines, neuroinflammation, cytokines

## Abstract

Depression presents a significant global health burden, necessitating the search for effective and safe treatments. This investigation aims to assess the antidepressant effect of the hydroethanolic extract of *Anacardium occidentale* (AO) on depression-related behaviors in rats. The depression model involved 42 days of unpredictable chronic mild stress (UCMS) exposure and was assessed using the sucrose preference and the forced swimming (FST) test. Additionally, memory-related aspects were examined using the tests Y-maze and Morris water maze (MWM), following 21 days of treatment with varying doses of the AO extract (150, 300, and 450 mg/kg) and Imipramine (20 mg/kg), commencing on day 21. The monoamines (norepinephrine, serotonin, and dopamine), oxidative stress markers (MDA and SOD), and cytokines levels (IL-1β, IL-6, and TNF-α) within the brain were evaluated. Additionally, the concentration of blood corticosterone was measured. Treatment with AO significantly alleviated UCMS-induced and depressive-like behaviors in rats. This was evidenced by the ability of the extract to prevent further decreases in body mass, increase sucrose consumption, reduce immobility time in the test Forced Swimming, improve cognitive performance in both tests Y-maze and the Morris water maze by increasing the target quadrant dwelling time and spontaneous alternation percentage, and promote faster feeding behavior in the novelty-suppressed feeding test. It also decreased pro-inflammatory cytokines, corticosterone, and MDA levels, and increased monoamine levels and SOD activity. HPLC-MS analysis revealed the presence of triterpenoid compounds (ursolic acid, oleanolic acid, and lupane) and polyphenols (catechin quercetin and kaempferol). These results evidenced the antidepressant effects of the AO, which might involve corticosterone and monoaminergic regulation as antioxidant and anti-inflammatory activities.

## 1. Introduction 

Depression is one of the major psychiatric disorders with a complex and multifactorial etiology [1]. One in five people suffer or will suffer from depression in their lifetime [2]. Lima et al. [3] estimated that depression will become the leading cause of morbidity worldwide. Depression strikes unexpectedly and, in its various forms, approximately 322 million individuals in 2017, constituting roughly 5% of the global population. Its prevalence varied across regions, with rates of 16% in the Eastern Mediterranean region, 9% in the African region, 12% in the European region, 15% in the region of the Americas, 21% in the Western Pacific region, and 27% in the South East Asia region [4]. 

The main causes of depression remain stressful life situations [5]. Stress is a biological alarm and defense reaction of the body in case of aggression [6]. Although stress is a normal and beneficial reaction for the body, when chronic, however, it becomes deleterious for the brain [7]. In fact, chronic and repeated stressors lead to hypothalamic–pituitary–adrenal (HPA) axis dysregulation, increased cortisol secretion and proinflammatory cytokines, and decreased monoamine expression. Such dysfunctions are observed both in human and animal models of depression [8]. Although the availability of drugs like monoamine oxidase inhibitors (MAOIs), tricyclic antidepressants like imipramine, and selective serotonin reuptake inhibitors (SSRIs) [4,5,6,7,8,9], access to these medications is limited in many developing countries. Additionally, these drugs are associated with a multitude of side effects [4]. Based on these limitations, researchers are increasingly turning to new medical alternatives, such as phytotherapy, for the prevention and/or treatment of human pathologies [10]. *Anacardium occidentale* L. (AO), 1753 (Anacardiaceae) is commonly employed in Africa to address issues related to inflammation, hypertension, and diabetes [11]. Scientific research has demonstrated that extracts from the leaves of this plant exhibit various beneficial properties. These include antidiabetic effects [11], anti-inflammatory properties [12], and anti-hypertensive characteristics [13], as well as antioxidant capabilities [14]. It is worth noting that oxidative stress is a significant contributor to the development of depressive disorders [15,16]. Furthermore, Dharamveer et al. [17] scientifically demonstrated the antidepressant activity of AO leaves but without studying the signaling pathways involved in this biological activity. Based on this evidence and the wide consumption of cashew nuts in northern Cameroon, we hypothesized that it may be endowed with an antidepressant-like effect against UCMS-induced depression in rats. 

The UCMS model is a widely recognized preclinical research paradigm used to mimic depressive and anxiety-like behaviors in rodents via chronic exposure to mild psychosocial stressors [18]. This model has demonstrated face, predictive, and construct validity, making it one of the few models where chronic, rather than acute, monoaminergic antidepressant administration proves effective [19]. The UCMS protocol induces a depressive-like state in animals, akin to human depression characterized by apathy and anhedonia [19]. It evaluates stress responses and antidepressant effects via behavioral measures such as spontaneous motivation, spontaneous grooming behavior, and appetence for pleasurable food [19]. This model’s high degree of unpredictability and uncontrollability of stressors, along with its use of mild stressors, enhances its relevance to human conditions [18]. Importantly, the UCMS model offers a translational bridge for investigating the pathophysiology of depression and testing potential therapeutic pharmacological agents in rodents [18].

The overall aim of this study is to evaluate the potential antidepressant effects of cashew nuts and to identify the molecular pathways that underlie their pharmacological actions. 

## 2. Results and Discussion

### 2.1. Chemical Composition of A. occidentale Fruit Kernel

The phytochemical analysis of the hydroethanol extract of AO was carried out using high-performance liquid chromatography coupled mass spectrometry (HPLC-MS). The chromatograms obtained revealed the presence of several secondary metabolites. The most abundant in decreasing order were beta-sitosterol, a ubiquitous plant sterol, ursolic acid, oleanolic acid, lupane, catechins, and limonene. Different peaks in Figure 1 represent the spectra, while the names, structure, and formula are enclosed in Table 1.

### 2.2. In Vitro Antioxidant Activity of A. occidentale Fruit Kernel Extract 

Figure 2 shows the result of the in vitro evaluation of the hydroethanolic extract of the AO fruit kernel. The anti-radical potential of the extract is higher (69.32 mg Trolox equivalent/100 g dry mass) than its reducing potential (55.45 mg Trolox equivalent/100 g dry mass) (Figure 2). Moreover, the extract’s 50% inhibitory concentration was 150 μg/mL, while that of the standard was 100 μg/mL (Table 2).

Oleanolic acid and ursolic acid are two pentacyclic triterpenoids that exhibit significant biological efficacy in vitro and in vivo. In the brain, ursolic acid reduces the microglia and astrocytes activation, downregulates the expression of NOS and COX-2, and decreases the interleukins and TNFα expression. It also attenuates cognitive deficits and depression and inhibits the NF-κB nuclear signaling pathway in the prefrontal cortex [20,21]. On the other hand, oleanolic acid modulates astrocytic, microglial, and neuronal functions, thereby offering neuroprotection against inflammatory-induced damage [22,23]. James et al. [24] revealed the ability of oleanolic acid to inhibit monoamine oxidase in depressive conditions. Lupane or Lupeol is a dietary triterpene present as an active component in a variety of medicinal plants with strong antioxidant, anti-inflammatory, and antiarthritic via Ras signaling pathways and NF-κB downregulation. NF-κB plays a central role as a mediator in the stress-induced impairment of neurogenesis and the development of depressive behavior. Its blockade in both peripheral immune cells and the brain inhibits the actions of other cytokines promoting inflammation (IL-1, IL-6, and TNF-α), implicated in stress and depression [25]. Quercetin, a dietary flavonoid found in the extract, is recognized for its antidepressant properties, primarily achieved by restraining the monoamine oxidase-A (MAO-A) activity. In the brain, MAO-A is integral in regulating neurotransmitter 5-hydroxytryptamine (5-HT) metabolism [26,27,28]. In the treatment of depression, one of the therapeutic strategies involves inhibiting the A isoform [29,30]. In short, all of these secondary metabolites of AO extract with the addition of antioxidant, anti-inflammatory, nuclear transcription factor regulator, and/or monoamine regulator properties, could synergistically combine their biological effects in order to monitor the expression of mediators implicated in the onset of depression or in the maintenance of the depressive state of animals. 

### 2.3. Impact of the Hydroethanolic Extract of A. occidentale Fruit Kernel on the Body Mass of Rats Changes 

Figure 3 illustrates the impact of the hydroethanolic extract of *Anacardium occidentale* (AO) on rats’ body mass variation during the experiment. Exposing rats to stress significantly (*p* < 0.05) decreased their body mass compared to those who were not exposed to stress. Despite exposure to stress, the body mass of rats treated with imipramine (20 mg/kg) and with extract (150 mg, 300 mg, and 450 mg/kg) increased. However, this increment was non-significant compared with stressed and untreated animals. 

In rodents, exposure to acute stress of moderate to severe intensity can cause a reduction in food intake and/or weight gain [31]. Weight loss may be observed whether there is an increase in food intake [32] or not [33] because other factors may be associated with weight change during stress. It has been shown that the cortisol produced during stress can lead to an increase in energy expenditure, tissue fatty acid oxidation, and an increase in body temperature [34]. In most cases, this reduction in weight is not compensated for even after the stress has ceased, and animals that have been stressed retain their reduced weight compared with unstressed animals [35]. These findings indicate that the AO extract effectively counteracted the effects of stress by regulating body weight gain. In contrast, rats in the UCMS control group exhibited a continuous reduction in body weight. These results show that AO extract corrected the effects of stress by regulating body weight gain, in contrast to animals in the UCMS control group, who continued to experience a persistent reduction in body weight.

### 2.4. The Impact of the Hydroethanolic Extract of A. occidentale Fruit Kernel on Sucrose Consumption

Figure 4A reveals that prior to the beginning of the study, there were no significant differences in 1% sucrose intake among the groups. However, after 21 days, the group subjected to stress (negative control) exhibited a reduction (*p* < 0.001) in the sucrose consumption parameter compared to the unstressed control (Figure 4B). The administration of anhedonic rats with different doses of the extract effectively counteracted the impact of stress by increasing their sucrose intake (Figure 4C). Significance was observed within the 150 mg/kg (*p* < 0.01) and 450 mg/kg (*p* < 0.001) doses when compared to the negative control group. The antidepressant drug imipramine also significantly (*p* < 0.05) increased sucrose consumption compared to the negative control group.

### 2.5. The Impact of the Hydroethanolic Extract of A. occidentale Fruit Kernel on Sucrose Preference 

The results revealed that before the onset of stress, all rats had a sucrose preference greater than 65% (Figure 5A). After 21 days of stress exposure, their sucrose preference index decreased (63.62%) compared to rats that were not subjected to any stressors (91.91%), as displayed in Figure 5B. In contrast, the administration of AO extract at 150, 300, and 450 mg/kg to anhedonic rats significantly boosted their preference for sucrose, resulting in preference rates of 81.17%, 75.09%, and 83.45%, respectively. Imipramine exhibited a similar increase in sucrose preference effect to that observed with the 450 mg/kg dose of the extract, with a value of 83.36% (Figure 5C). Imipramine displayed a similar biological effect to the extract at the 450 mg/kg dose, resulting in an increase in sucrose preference to 83.36% (Figure 5C). 

Anhedonia can be considered a loss of interest or pleasure in rodents. This parameter is an indicator of the depressive state, as loss of pleasure is considered to be a key symptom of depression in humans. According to Liu et al. [36], an animal with a sucrose preference below 65% is considered anhedonic. This could be explained by dysfunction of dopamine neurotransmission in the mesolimbic region [3]. The mesolimbic region of the brain encompasses dopamine neurons located within the ventral tegmental area and their projections into the forebrain, notably involving medium spiny neurons within the nucleus accumbens. The mesolimbic dopamine system is linked with the rewarding effects of food, sex, abuse of drugs, and anhedonia, as well as the decrease in motivation and energy levels observed in patients suffering from depression. It thus contributes more importantly to the etiology, pathophysiology, and symptomatology of depression [37]. In the current study, exposure to unpredictable chronic mild stress for 42 days resulted in a significant reduction in sucrose preference among all rats subjected to this unpleasant experiment. Treatment with the extract inhibited the anhedonia-related behaviors in the rats and significantly sustained their sucrose preference at levels significantly surpassing those observed in the UCMS control group, reaching a range considered normal. Other findings revealed that extracts endowed with the ability to increase sucrose preference could have antidepressant properties [1,37,38,39]. These findings strongly indicate that the hydroethanolic extract of AO may possess antidepressant activity via its potential regulation of the mesolimbic dopamine system in the brain. To corroborate these findings, additional behavioral tests, including the forced swimming test (FST), the novelty-suppressed feeding test (NSFT), and an analysis of monoamine concentrations in the mesolimbic systems were evaluated.

### 2.6. Impact of the Hydroethanolic Extract of A. occidentale Fruit Kernel on Immobility Time, Swimming, and Climbing in the Forced Swimming Test 

The forced swimming test (FST) results (Figure 6) provide valuable insights into the impact of the hydroethanolic extract of AO on the behavior of rats subjected to chronic mild unpredictable stress (UCMS). The FST is a commonly used behavioral test for assessing depressive-like behaviors and the potential antidepressant effects of substances. 

Swimming time is a parameter that measures the time that rats spent actively swimming, and is considered an active escape behavior in the FST or a measure of struggling behavior. In our study, UCMS exposure led to a significant reduction in swimming behavior compared to non-exposed rats, which indicates a depressive-like state (*p* < 0.001) (Figure 6A). This result aligns with the expectation that animals subjected to chronic stress are more likely to display passive and despairing behaviors. However, treatment of anhedonic rats with the hydroethanolic extract of AO, as well as imipramine, increased swimming time. We observe that the lower dose (150 mg/kg) led to a more pronounced rise in swimming time compared to the 300 and 400 mg/kg doses. This effect may be linked to the biphasic dose–response characteristics of the hydroethanolic extract of AO [40,41,42,43,44]. Our results suggest that both AO and imipramine have an antidepressant-like effect by promoting active coping strategies in response to the stressor.

Climbing time in the FST is another active response, representing the duration rats spent trying to climb out of the water and reflecting the animal’s efforts to escape the stressful condition. The findings demonstrate that rats exposed to UCMS had shorter climbing times compared to non-exposed rats, indicating a significantly diminished capacity to engage in active escape attempts (*p* < 0.01) (Figure 6B). Treatment of anhedonic rats with AO extract and imipramine led to a significant rise in climbing behavior (*p* < 0.001). Similar to time to swim, both AO and imipramine promote active and goal-directed behaviors in response to the stressor, indicating a reduction in depressive-like behaviors.

While immobility in the FST reflects a passive and despairing state, UCMS exposure significantly augmented the immobility time (*p* < 0.001) (Figure 6C) when compared with non-exposed rats, which is generally associated with depressive-like behavior. However, all doses of 150 mg/kg, 300 mg/kg, and 450 mg/kg of the AO extract, along with imipramine (20 mg/kg), demonstrated a significant reduction (*p* < 0.001) in immobility time when contrasted with the UCMS control group. The reduction in immobility time implies an antidepressant-like effect as the rats become less passive and more involved in active behaviors. In addition, a dose–response effect was observed, with a higher decrease recorded for the lower dose (15 mg/mg) followed by the doses 300 mg/kg and 400 mg/kg. Similar to swimming time results, this pattern can be attributed to the biphasic dose–response properties of the hydroethanolic extract of AO [40,41,42,43,44]. 

Taken together, the results from the FST indicate that UCMS exposure induces depressive-like behaviors characterized by increased immobility and reduced swimming and climbing. The treatment with the AO extract at all doses and the reference drug imipramine led to significant improvements in the animals’ behavior. These improvements include increased active escape behaviors (swimming and climbing) and reduced passive despair (immobility).

These findings suggest that the AO hydroethanolic extract possesses antidepressant-like effects, as it was able to reverse the depressive behaviors induced by chronic stress. The FST results align with the earlier findings of the sucrose preference test and further support the potential antidepressant effects of this extract.

### 2.7. Impact of the Hydroethanolic Extract of A. occidentale Fruit Kernel on the Latency to Consume the Food in the Novelty-Suppressed Feeding Test

In our observations, rats in the stressed groups took significantly (*p* < 0.001) longer time to consume their food than the rats in the unstressed group (Figure 7). Furthermore, a significant decrease (*p* < 0.001) in food consumption latency was recorded in the extract-treated rats to that of the UCMS control group.

The elevation in food consumption latency in the negative control rats compared with the unstressed group in the NSFT implies that stress induces a reduced appetite. This decrease in appetite may be attributed to potential alterations in serotonin transmission within the brains of stressed rats [39]. In addition, after the extract treatment and whatever the dose used, a significant decrease in latency was noted, in contrast to the negative control group. The NSFT serves as a valuable tool for investigating the antidepressant potential of drugs and uncovering insights into their mechanisms of action. This test is sensitive to antidepressants in chronic and subchronic administration but not in acute administration. Many stress-induced animal models of depression, including UCMS and chronic corticosterone administration, often lead to an extended latency to eat during the novelty-suppressed feeding test. This delay in eating behavior can be alleviated via chronic treatment with antidepressant medications [39]. When the treatment is positive, as is the case with hydroethanolic extract of *A. occidentale* fruit kernel, the reduction in latency time is correlated with a rise in the proliferation of adult neural progenitor cells in the dentate gyrus of the hippocampus. Prior research has indicated that the ablation of this niche of adult neurogenesis results in the absence of the antidepressant-induced reduction in latency to eat during the novelty-suppressed feeding test [45]. This implies that adult neurogenesis is necessary to facilitate the positive impacts of the plant extract. Furthermore, exposure to UCMS elevated the immobility time in comparison to the unstressed group in the FST. Immobility is an indicator of desperation [37]. It is thought to be due to a decrease in serotonin and norepinephrine stores in rats’ brains [8]. Chronic administration of the hydroethanolic extract of AO nuts increased swimming and climbing time at the expense of immobility. These results are similar to some previous studies [37,45]. The group that received imipramine (20 mg/kg) likewise reduced the latency time in the NSFT and the immobility time in the FST when compared to the UCMS control group. These results from NSFT and FST collectively indicate that the hydroethanolic extract from AO nuts effectively counteracts depressive behaviors in anhedonic rats.

### 2.8. Effect of the Hydroethanolic Extract of A. occidentale Fruit Kernel in the Morris Water Maze Test

According to the results in Figure 8, a statistically significant increase in the time required to find the hidden platform and a decrease in time spent in the target quadrant was observed in batches of stressed rats compared with the unstressed control group. In addition, a significant reduction in latency to escape from the platform was recorded in batches of rats treated with hydroethanol extract of AO at 150 mg/kg and 300 mg/kg. These changes indicate the attenuation of stress effects. A similar reduction in latency to localize the escape platform (*p* < 0.05) and a rise in time spent in the target quadrant were recorded by imipramine (20 mg/kg). In addition, analysis of the rats’ heat diagrams and pool travel times showed that rats in the negative control group spent less time in the target quadrant than the unstressed group. In comparison, extract-treated rats spent more time in the target quadrant, as shown by the heat diagram and the length of trips in the maze (Figure 8C).

### 2.9. Impact of A. occidentale Fruit Kernel Hydroethanolic Extract on Memory Impairment in the Y-Maze Test

Figure 9 shows the potential impact of AO extract on the percentage of spontaneous alternations in the Y-maze. A significant decrease in the percentage of spontaneous alternations was noted in the UCMS control group compared with the unstressed group (*p* < 0.001). In contrast to the negative control group, supplementation with extracted AO in the 300 and 450 mg/kg rat models showed a significant improvement in the percentage of spontaneous alternation (*p* < 0.001). Imipramine (20 mg/kg) showed a similar profile.

Many patients with depression exhibit memory deficits [46]. Cognitive deficits are recognized as a hallmark of depressive disorders in accordance with the *Diagnostic and Statistical Manual of Mental Disorders* (DSM-V) [47]. Stress is known to have the capacity to impede cognitive functions, including learning and memory, and this impact is associated with elevated glucocorticoid levels [48,49]. Memory impairment has been evaluated in numerous works using stress models. In our current investigation, we assessed memory dysfunction via stress model paradigms. Our findings reveal stress-induced memory impairment, affecting both short-term and long-term memory. This was confirmed via the utilization of the Y-Maze for short-term memory assessment and the Morris water maze (MWM) for long-term memory assessment. The outcomes indicated that in the control group exposed to stressors, there was an extended duration required to find the escape platform, and less time was spent in the specific location where the platform was positioned. The MWM test has gained significance in assessing memory in stress models. It relies on reduced escape time and decreased time spent in the designated quadrant as reliable indicators of memory impairment [50,51,52]. These effects are believed to be consequences of serotonin deficiency [36], inflammation, and/or oxidative stress following cortisol overproduction [53]. The administration of the extract effectively rectified these memory deficits by decreasing the time needed to find the platform and enhancing the allocated time to the target quadrant. In the Y-Maze assessment, the extract improved spontaneous alternation in rats compared to stressed rats that did not receive any treatment. These results suggest an enhancement in memory, potentially attributable to the improvement in the bioavailability of serotonin on one hand and the capacity of the extract to mitigate oxidative stress [54] and inflammation following hypercortisolemia during stress on the other hand. 

### 2.10. Impact of the A. occidentale Fruit Kernel Hydroethanolic Extract on Serum Corticosterone Levels 

The impact of the extract on corticosterone levels is illustrated in Figure 10. In contrast to the unstressed group, the negative control group recorded a significant increase (*p* < 0.001) in corticosterone levels. Interestingly, in contrast to the negative control group, all extract doses (150, 300, and 450 mg/kg PC) showed a decrease in corticosterone levels.

During stress, the hypothalamic–pituitary–adrenal (HPA) axis is activated via the CRF. Via the production of numerous molecules, notably glucocorticoids, this axis is recognized as one of the major regulators of many chronic pathologies, such as depression [55]. Cortisol in humans and corticosterone in rodents produced during stress is for adaptive purposes [56]. If poorly managed, this stress could lead to elevated blood levels of this hormone. Exposure to high levels of corticosterone causes damage to the brain, mainly the hippocampus, one of the brain regions where glucocorticoid receptors are highly concentrated and induce symptoms like cognitive decline and depression. Several studies on stress models also reported an excessive amount of corticosterone in the blood of stressed animals [57,58]. This increment is thought to be due to a dysregulation of the HPA axis via a defect in negative feedback control of cortisol [59]. In this study, chronic stress induces a significant increase in blood corticosterone levels and depression-like behavior. However, this effect was effectively regulated via the repeated administration of the hydroethanolic extract derived from AO fruit kernels. Corticosterone is a primary target of substances that reduce chronic stress, reducing its cerebral concentration, as observed in our findings, and serves as an indicator of the antidepressant activity of our plant extract.

### 2.11. Effect of the Hydroethanolic Extract of A. occidentale Fruit Kernel on Brain Pro-inflammatory Cytokines Levels

Figure 11 displays the impact of the hydroethanolic extract from AO fruit kernel on the cerebral levels of three pro-inflammatory cytokines: tumor necrosis factor (A), interleukin-1β (B), and interleukin-6 (C). Notably, in the negative control rats, the levels of the tumor necrosis factor, interleukin-1β, and interleukin-6 were significantly elevated (*p* < 0.001) compared to those in the unstressed rats. Conversely, and in contrast with the UCMS—rat group, extract treatment of rats led to a significant reduction (*p* < 0.001) in the tumor necrosis factor and interleukin-1β levels. Furthermore, even the lowest dose of the plant extract (150 mg/kg) was able to significantly (*p* < 0.001) decrease interleukin-6 expression, similar to the effect of imipramine.

Chronic exposure to stress, particularly chronic mild stress, is associated with a melancholic form of depression characterized by heightened insulin resistance and elevated levels of pro-inflammatory cytokines. In these cases, the immune system becomes adaptively activated due to the excessive release of cortisol. Recent data have accumulated substantial evidence pointing to the involvement of cytokines in the pathogenesis of chronic depression. Cytokines serve as signaling molecules that regulate the immune system and can either trigger or inhibit inflammatory processes. Pro-inflammatory cytokines can breach the blood–brain barrier or are generated by overactive neural cells, thus disrupting neurotransmission. Depressed patients have been found to display heightened levels of pro-inflammatory cytokines, such as TNF-α, IL-1β, IL-2, IL-6, and IL-12. TNF-α, which is released by activated macrophages, promotes cellular apoptosis and the extracellular release of damage-associated molecular patterns (DAMPs). This, in turn, activates macrophages and microglial cells to release interleukins (IL-1β, IL-6, and IL-23), chemokines, and reactive nitrogen and oxygen species while also upregulating NF-κB expression [60,61,62]. In this study, rats exposed to UCMS showed elevated plasma proinflammatory cytokines compared with the unstressed rats in this study. A peripheral increase in circulating cytokines may directly activate brain microglia via the nerve or by passively crossing the blood–brain barrier [63]. These cytokines can also hijack serotonin biosynthesis via activation of Indoleamine 2,3-dioxygenase, which alters monoamine biosynthesis by degrading tryptophan into Kynurenine [64,65]. Treatment with the extract led to a significant reduction in the concentrations of interleukin-1β, TNF-α, and interleukin-6 in the plasma of rats compared to stressed rats that did not receive any treatment. The decrease in cytokines could confer a central anti-inflammatory effect on the extract, resulting in better bioavailability of monoamines and neuronal cell survival.

### 2.12. Effect of the Hydroethanolic Extract of A. occidentale Fruit Kernel on Brain Monoamines Levels 

A statistically significant decrease in monoamine levels was observed in the stressed batches compared to the unstressed group (Figure 12). A correction of the deficit marked by the significant increase in serotonin, norepinephrine, and dopamine was observed after treatment with AO extract at all doses (150 mg/kg, 300 mg/kg, and 450 mg/kg) compared with the non-stressed group. Similar results were obtained with the imipramine group compared with the UCMS control group.

The monoaminergic hypothesis of depression posits that the underlying pathophysiology of depression is connected to a reduction in serotonin, norepinephrine, and/or dopamine levels in the central nervous system [3,66]. It is also noteworthy that current conventional antidepressants enhance monoamine neurotransmission [67]. Numerous studies have demonstrated that chronic stress patterns lead to a decrease in norepinephrine, serotonin, and dopamine levels [45,68,69]. In this study, stress induced a significant reduction in norepinephrine, serotonin, and dopamine levels in brain homogenates of stressed animals. All doses of hydroethanolic extract of AO nuts significantly increased the concentration of these monoamines when compared to the UCMS control group. Shen et al. [37] demonstrated that the antidepressant effects of an extract could be associated with its capacity to enhance the bioavailability of monoamines in deficient brain areas. These results confirm the antidepressant action of the extract of AO obtained during the behavioral tests but are also in line with the central anti-inflammatory potential of the extract noted with the cytokines.

### 2.13. Impact of A. occidentale Fruit Kernel Hydroethanolic Extract on Lipid Peroxidation

Figure 13 illustrates the impact of the extract on MDA concentration. The negative control group’s MDA concentration was significantly higher (*p* < 0.001) compared to that of the unstressed group. In contrast, the extract-treated groups exhibited a significantly lower (*p* < 0.001) MDA concentration in brain homogenates.

### 2.14. Impact of the Hydroethanolic Extract of A. occidentale Fruit Kernel on SOD Activity 

The stressed group showed significantly lower levels of SOD than the control (*p* < 0.001). A correction of the antioxidant defense deficit marked by an increase in the antioxidant enzyme SOD was observed after treatment with hydroethanolic extract of AO fruit kernel (Figure 14).

Oxidative stress is a physiological condition resulting from an imbalance between reactive oxygen species (ROS) and the body’s antioxidant system [70]. A poor antioxidant defense, high concentration of polyunsaturated fatty acids, and high oxygen utilization by the brain make it particularly vulnerable to oxidative damage [71]. Based on all the above, it is known that frequent exposure of nerve cells to glucocorticoids could induce peroxidation of membrane lipids and a decrease in the antioxidant defense of the brain during stress [49,53] and is the reason why we opted to verify the potential of the extract to modulate oxidative stress markers in this study. MDA, a product of lipid peroxidation, was greater in expression with exposure to UCMS, especially in the negative control group [72,73]. The administration of AO extract and imipramine reversed this trend by decreasing MDA levels and increasing the antioxidant enzymes (SOD and catalase) activity in brain homogenates. SOD and catalase are responsible for reducing superoxide radicals to hydrogen peroxide and hydrogen peroxide to H_2_O, respectively [70]. The elevation in antioxidative capacity and the reduction in the MDA level suggested an antioxidant effect of the extract. These findings corroborate the effects observed in studies of the antioxidant activity in vitro of the extract and further support the neuroprotective potential of this extract. It is known that oxidative stress is involved in neuronal loss [70], one of the pathophysiological mechanisms of depression. An improvement in oxidative status would be a pathway by which the extract exerts its antidepressant effects.

### 2.15. Impact of the Hydroethanolic Extract of A. occidentale Fruit Kernel on Hippocampal and Prefrontal Cortex Neurons

Figure 15 below illustrates the effects of treatment with the hydroethanolic extract of AO on the structural organization of the dentate gyrus, CA1, CA3, and prefrontal cortex neurons. In the unstressed group, the cellular layers remained intact, and there was a highly organized arrangement of neurons in the prefrontal cortex, pyramidal layer, and dentate gyrus of the hippocampus. However, in the negative control rats, there was a noticeable neuronal loss in the cortex and dentate gyrus. In contrast, the histology of extract-treated rats revealed a normal structural organization of neurons in the hippocampus and cortex. 

Based on the results of histological sections, an almost normal architecture of zone CA3 of the hippocampus and prefrontal cortex was observed in batches of rats treated with e hydroethanolic extract of *A. occidentale*. Whereas the batches of rats subjected to UCMS showed cellular losses in these areas. These cellular alterations due to the deleterious effects of glucocorticoids could also affect learning and memory functions [71], as the hippocampus and prefrontal cortex play a very important role in cognitive processes.

## 3. Material and Methods

### 3.1. Chemical Products 

All the chemicals used in the various investigations were purchased from the following companies: BDH Chemicals Ltd. (Poole, UK), Sigma-Aldrich (St. Louis, MO, USA), and Shanghai Biochemical Co., Ltd. (Shanghai, China). 

### 3.2. Plant Material 

Samples of *Anacardium occidentale* Linné (AO) 1753 nuts were collected in March 2021 in Maroua, in the Far North of Cameroon. After taxonomic identification, specimens (Voucher n°41935/HNC) were deposited at the Herbier National du Cameroun (HNC, Yaoundé) and authenticated via comparison with material deposited by Sétabié and Letonzey (Voucher n°41935/HNC).

#### 3.2.1. Extract Preparation 

The cashew nuts were cracked and opened with a knife to release the kernel (1450 g). These kernels were then dried in a well-ventilated environment at room temperature and in the shade. Once dried, the kernels were reduced to a paste weighing 1385 g. The samples were then macerated for 72 h in ethanol–water (80/20, *v*/*v*) solvent. After maceration, the mixture was filtered through filter paper No. 4. The filtrate obtained was then concentrated at 60 °C using a rotavapor to remove the ethanol. The resulting extract was freeze-dried at −55 °C.

#### 3.2.2. Assessment of the In Vitro Antioxidant Potential 

##### Evaluation of the Reducing Activity of Fe^3+^ to Fe^2+^ of the Extract 

The extract’s reducing capacity was assessed by determining its ability to convert iron (Fe^3+^) to Fe^2+^ using the method described by Suresh et al. [72]. 

##### Evaluation of the Anti-Radical Activity of the Extract

The extract’s anti-radical activity was assessed following the method described by Kansci et al. [73]. 

### 3.3. Experimental Animal

Forty-two adult male Wistar rat mice with an initial body weight of 100–166 g were used. The rats were supplied by the Biophysics and Biochemistry Laboratory, Food Science and Nutrition Department, University of Ngaoundéré, Cameroon. They were transferred to the animal house of the Laboratory of Animal Physiology and Pharmacognosy, University of Maroua. They were placed in a controlled environment two weeks before the start of the experiment (T = 25 ± 2 °C, and 12 h light/dark cycles, with standard rat food and water ad libitum). The experiment was conducted in accordance with the “Guide pour le soin et l’utilisation des animaux de laboratoire” manual (8th Edition), in compliance with the guidelines of the Cameroon Bioethics Committee (reg N° FWAIRB00001954) and approved by the Bioethics Advisory Commission of the Faculty of Sciences, University of Maroua (ref: No14/0261/Uma/D/FS/VD-RC). 

### 3.4. Unpredictable Chronic Mild Stress and Distribution of Animals 

The method used was described by Duccoted et al. [74] with slight modifications via the addition or elimination of some stressors and modulation of their duration. Forty-two rats were divided into two groups: a normal group of 7 rats and a stressed group of 35 rats. The second (stressed) group was subjected to the unpredictable chronic mild stress (UCMS) for 21 days. After these 21 days, the sucrose preference test and the novelty-suppressed feeding test were performed. Only rats with a sucrose preference less or equal to 65% (anhedonic rats) were selected for further experimentation. The stress battery consisted of eight stressors (water deprivation, food deprivation, sound stimulation + night illumination, wet bedding, isolation, physical restriction, forced swimming at 30 °C, and no stress). The timing and duration of stressors were varied daily to minimize their predictability (Table 3). Anhedonic rats were randomized into five (5) groups of six (6) rats each: negative control group (distilled water + UCMS), positive control group (UCMS + Imipramine), and three test groups receiving extract doses (UCMS + 150 mg/kg, UCMS + 300 mg/kg, and UCMS + 450 mg/kg) for an additional 21 days.

### 3.5. Behavioral Tests

#### 3.5.1. Sucrose Preference Test (SPT)

The SPT can be used to show whether rats are in an anhedonic state (characterized by a significant decrease in sucrose intake or not) [75]. The SPT was performed before, mid-way, and at the end of the UCMS protocol. Without the deprivation of food or water, rats were given free choice for 24 h between two bottles, one containing a 1% sucrose solution and the other containing tap water. The following day, the bottle position was interchanged, and rats were once more allowed to freely drink from a bottle of their choice for 24 h. After the test, the bottles were collected and weighed, and the changes in the volume of their content were determined to calculate the percentage of sucrose preference according to the formula:$$Sucrose\;Preference\;(\%)=100\times \frac{Quantity\;of\;Sucrose\;\%\;(g)}{Quantity\;of\;Sucrose\;1\%\;\left(g\right)+Quantity\;of\;Water\;(g)}$$

#### 3.5.2. Forced Swimming Test (FST)

The forced swim test (FST), commonly used to study antidepressant activity [24], was conducted following the method described by Porsolt [76]. 

#### 3.5.3. Novelty Suppressed Feeding Test (NSFT) 

The protocol employed in this study is similar to that described by Kristen et al. [77]. 

#### 3.5.4. Y-Maze Test 

The Y-maze apparatus used in the study was designed with a chamber featuring three arms, which were labeled A, B, and C. These arms were symmetrically separated at 120 degrees from each other. Each mouse was initially placed in arm A of the Y-maze apparatus and allowed to explore all three arms for a period of 5 min. After each animal’s passage through the maze, it was cleaned with 70% ethanol to remove any residual odors. In the Y-maze test, the frequency of arm entries and alternations was assessed. Alternations referred to consecutive navigations through the three arms in the pattern of ABC, CAB, or BCA, but not BAB or ABA [78]. The number of maximum spontaneous alternation behavior was determined by subtracting 2 from the total number of arms entered. To assess memory performance, the percentage (%) of correct alternations, which serves as an index of memory performance, was calculated using the following formula: $$Spontaneous\;Alternation\;(\%)=100\times \frac{Total\;Number\;of\;Alternations}{Maximal\;Number\;of\;Alternations}$$

#### 3.5.5. Morris Water Maze (MWM)

The apparatus used in this study was a metallic cylinder with a diameter measuring 159 cm. It was divided into four quadrants: north, south, east, and west. The testing procedure was conducted over 8 days, following the method employed by Ngatanko et al. [79]. 

### 3.6. Sample Preparation and Biochemical Assay

#### 3.6.1. Sample Preparation

After the euthanasia of the mice on day 43 using a combination of intraperitoneal (i.p.) diazepam/ketamine at doses of 10 mg/kg and 50 mg/kg, respectively, blood and brain samples were taken. The brain was divided into two hemispheres. The right hemisphere was preserved in formalin (10%) for histological studies, while the left hemisphere was weighed, ground, and homogenized in phosphate buffer (0.1 mM, pH 7.0). The homogenate was centrifuged at 3000 rpm for 15 min at 4 °C, and the supernatants were collected. Levels of MDA, SOD, monoamines, and pro-inflammatory cytokines were measured. Blood samples were collected in dry tubes and centrifuged at 3000 rpm for 15 min. The serum obtained was used to determine corticosterone concentration.

#### 3.6.2. Biochemical Assay

##### Determination of Monoamine Levels

Determination of serotonin levels

The estimation of serotonin concentration was carried out in accordance with the method described by Robinson et al. [80]. 

2.Determination of norepinephrine levels

A volume of 0.2 mL supernatant was mixed with 0.05 mL HCl (0.4 M), 0.1 mL EDTA/sodium acetate buffer (pH 6.9), and 0.1 mL iodine solution (0.1 M in ethanol) for oxidation. After 2 min, 0.1 mL Na_2_SO_3_ solution was added to stop the reaction. After 15 min incubation, 0.1 mL acetic acid was added. The solution was then heated to 100 °C for 6 min and cooled to room temperature. The excitation and emission spectra were read using a spectrofluorometer calibrated at 395–485 nm. The concentration of noradrenaline (NA) was determined fluorometrically. This fluorescence is proportional to the level of NA read at a wavelength of 250 nm. A noradrenaline standard solution was prepared from different concentrations of noradrenaline (0, 3, 6, 8, 10, and 12 ng/mg) in distilled water and butanol/HCl (1/2). The concentration of noradrenaline was determined by the calibration curve according to the equation Y = AX + B [80].

3.Determination of dopamine levels

To quantify dopamine levels, 1 mL of homogenate was transferred into test tubes that contained 2.5 mL of heptane and 0.31 mL of a 0.1 M hydrochloric acid solution. After vigorously vortexing the mixture for 10 min, each tube was subjected to centrifugation at 2000 rpm for a duration of 10 min. All these procedural steps were executed at a temperature of 0 °C. The absorbance was assessed at 485 nm. The concentration of dopamine was determined by referencing a calibration curve via the equation Y = AX + B and was expressed in units of ng/mL [81].

##### Determination of Oxidative Stress Markers 

Determination of lipid peroxidation

The final product of lipid peroxidation, known as MDA, was quantified using the methodology outlined by Wills [82]. 

2.Determination of superoxide dismutase activity

For the determination of SOD activity, we followed the protocol of Misra and Fridovish [83].

##### Determination of Corticosterone Levels in Serum

Corticosterone levels were determined using an ELISA kit (CUSABI, Catalog Number. CSB-E07014r) in accordance with the manufacturer’s guidelines. 

##### Determination of Pro-Inflammatory Cytokines

Determination of Interleukin-6 (IL-6) levels

The IL-6 concentration was determined using an ELISA kit (CUSABI, Catalog Number. CSB-E04640r) in accordance with the manufacturer’s guidelines.

2.Determination of Tumor Necrosis Factor-α levels

TNF-α levels were determined using an ELISA kit (Catalog Number. CSB-E11987r) in accordance with the manufacturer’s guidelines.

3.Determination of Interleukin-1β levels

IL-1β levels were determined using an ELISA kit (Catalog Number. CSB-E08055) in accordance with the manufacturer’s guidelines.

### 3.7. Histopathological Studies

Histology was performed according to an atlas. Brain coronal sections (50 µm) of the hippocampal and prefrontal cortex regions were made. These sections were processed in increasing concentrations of ethanol and stained with Hematoxylin/eosin. Sections obtained were photographed (50×) using an optical microscope (Scientico, Haryana—India) equipped with a camera. The experimental design for the whole experiment is presented in Figure 16. 

### 3.8. Statistical Analysis

Statistical analysis was carried out using Graph Pad Prism software version 8.0.1 (San Diego, CA, USA). For univariate tests, including the forced swim test, novelty-suppressed feeding test, Morris water maze, Y-maze test (spontaneous alternation percentage), and all biochemical parameters, a one-way analysis of variance was conducted, followed by Tukey’s post hoc test to analyze the effects of treatments. In the case of the anhedonia test and the latency to find the platform in the Morris water maze test, a two-way ANOVA was applied, followed by the Bonferroni post hoc test. All data were presented as mean ± S.E.M per group, and statistical significance was considered when the *p*-value was less than 0.05.

## 4. Conclusions

In this study, the hydroethanolic extract of *Anacardium occidentale* (AO) kernel reversed UCMS-induced depressive and cognitive disorders. This effect could be related to the presence of triterpenoid compounds (ursolic acid, oleanolic acid, and lupane) and polyphenols (catechin quercetin and kaempferol) that could act via their antioxidant/inflammatory potential and their capacity to regulate corticosterone production and monoamines (dopamine, serotonin, and noradrenaline) neurotransmission.

## Figures and Tables

**Figure 1 brainsci-13-01561-f001:**
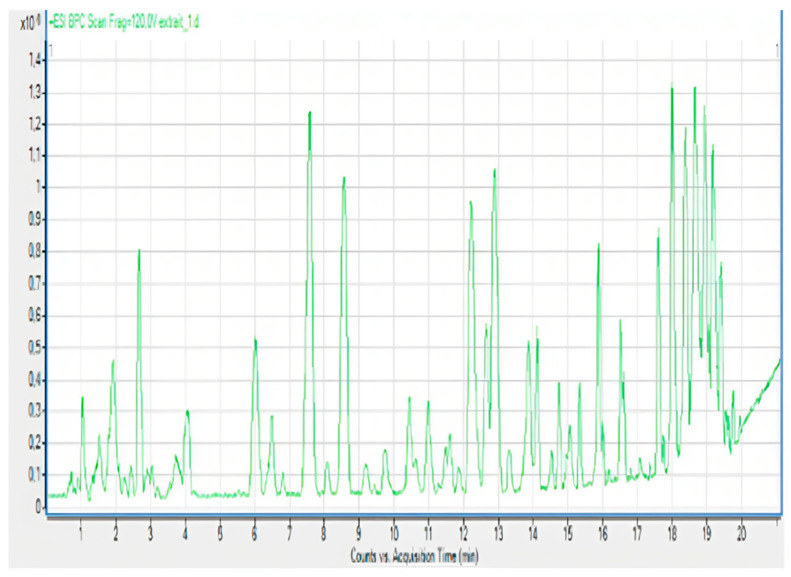
Chromatographic profile of *A. occidentale* fruit kernel extract. The numbers 1 to 11 represent the spectra of the identified compounds.

**Figure 2 brainsci-13-01561-f002:**
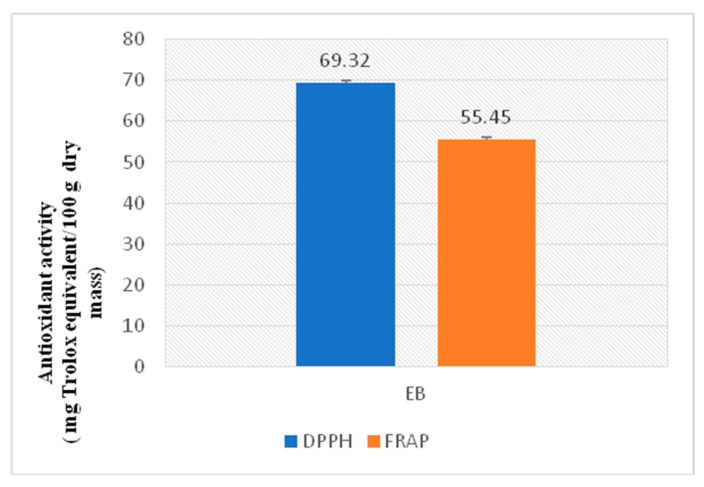
The in vitro antioxidant potential of the *A. occidentale* fruit kernel extract.

**Figure 3 brainsci-13-01561-f003:**
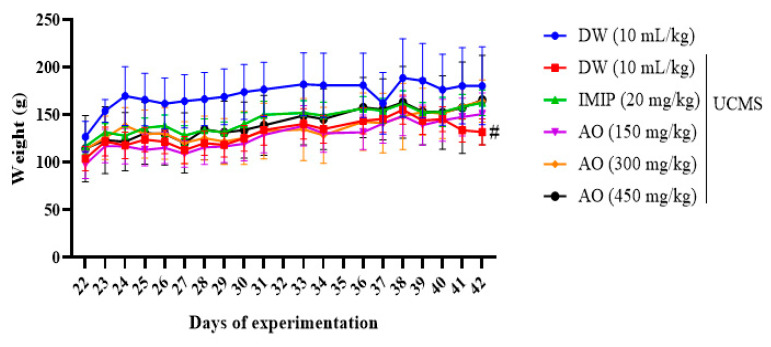
The impact of the hydroethanolic extract of *A. occidentale* fruit kernel on the body mass of rats exhibiting anhedonia. Each point indicates the mean ± SD; sample size, n = 6; DW = distilled water; IMIP = imipramine; AO = *Anacardium occidentale*; UCMS = unpredictable chronic mild stress. ^#^
*p* < 0.05 significant difference when compared to the unstressed group.

**Figure 4 brainsci-13-01561-f004:**
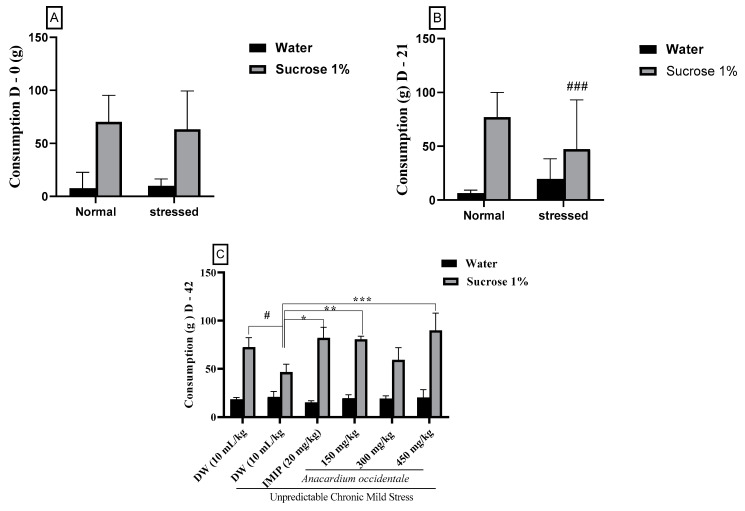
The impact of *A. occidentale* fruit kernel hydroethanolic extract on sucrose consumption in anhedonic rats. (**A**) Sucrose Consumption in Anhedonic Rats on Day 0; (**B**) Sucrose Consumption in Anhedonic Rats on Day 21; (**C**) Effects of *A. occidentale* Extract on Sucrose Consumption in Anhedonic Rats on Day 42; Each bar indicates the mean value ± SD, with a sample size of n = 5; DW = distilled water; IMIP = imipramine; AO = *Anacardium occidentale*; D = day. ^###^
*p* < 0.001; ^#^
*p* < 0.05 = significant difference compared to the unstressed group; * *p* < 0.05, ** *p* < 0.01, *** *p* < 0.001 = significant difference compared to the UCMS control group.

**Figure 5 brainsci-13-01561-f005:**
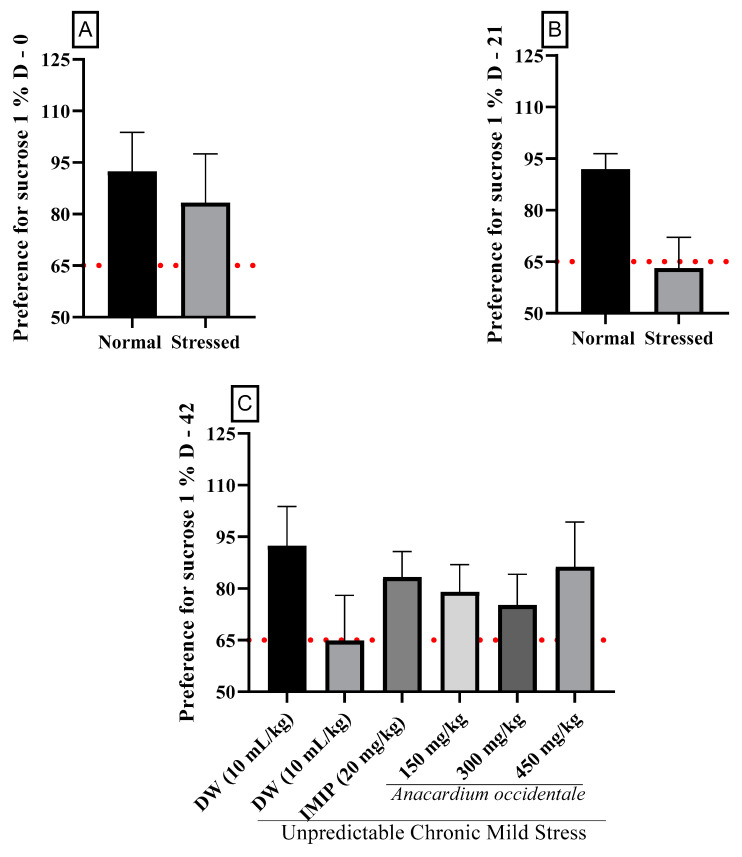
The impact of chronic mild unpredictable stress on sucrose preference in normal rats. (**A**) Baseline Sucrose Preference (Day 0); (**B**) Effect of Chronic Stress on Sucrose Preference (21 Days); (**C**) Effects of AO Extract and Imipramine on Sucrose Preference (Day 42) Each bar indicates the mean value ± SD, with a sample size of n = 5; DW = distilled water; IMIP = imipramine; AO = *Anacardium occidentale*; D = day.

**Figure 6 brainsci-13-01561-f006:**
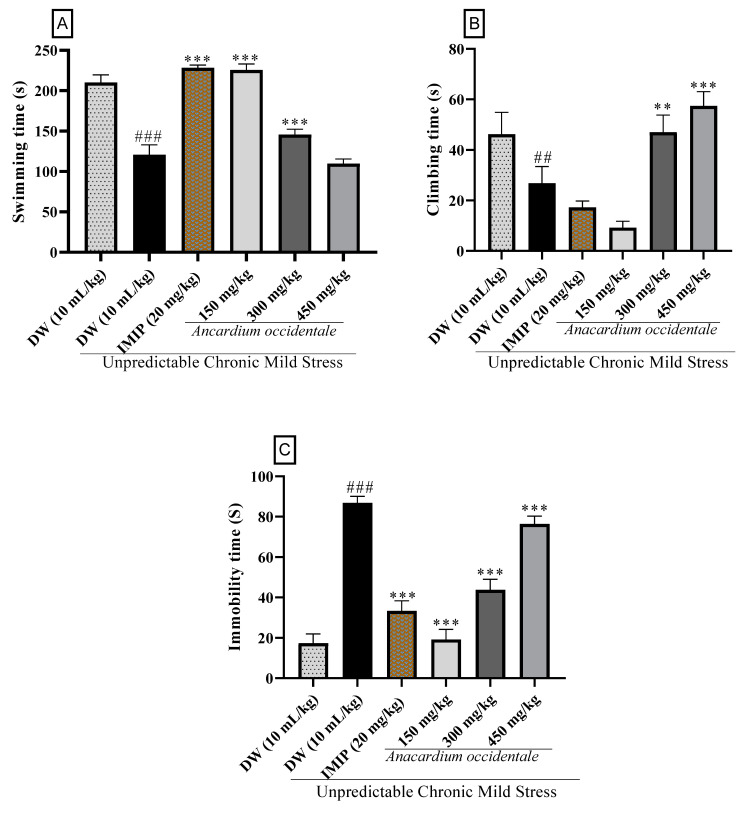
The impact of the hydroethanolic extract of AO fruit kernel on swimming, climbing, and immobility time in anhedonic rats during the forced swimming test. (**A**): “Forced Swimming Test—Swimming Time”; (**B**): “Forced Swimming Test—Climbing Time”; (**C**) “Forced Swimming Test—Immobility Time”. Each bar in the figure indicates the mean ± SD, with a sample size of n = 5; DW = distilled water; IMIP = imipramine; AO = *Anacardium occidentale*; ^##^ *p* < 0.01, ^###^ *p* < 0.001 significant difference compared to the unstressed group; ** *p* < 0.01, *** *p* < 0.001 significant difference compared to the UCMS control group.

**Figure 7 brainsci-13-01561-f007:**
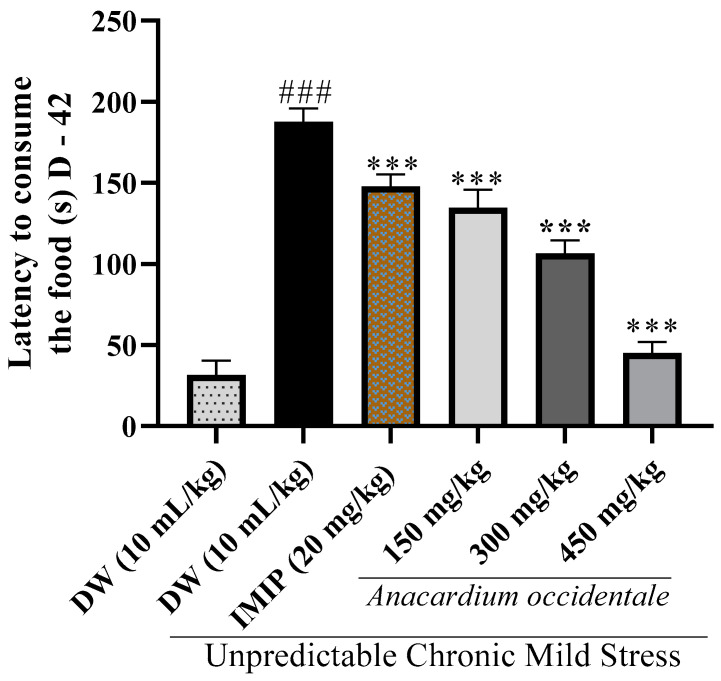
Behavioral results of the novelty-suppressed feeding test, presenting relative latencies until the first bite in Imipramine, *A. occidentale* treated and Vehicle control. Each bar indicates the average ±SD; n = 5; DW = distilled water; IMIP = imipramine; ^###^
*p* < 0.001 significant difference compared to the unstressed group; *** *p* < 0.001 significant difference compared to the UCMS control group.

**Figure 8 brainsci-13-01561-f008:**
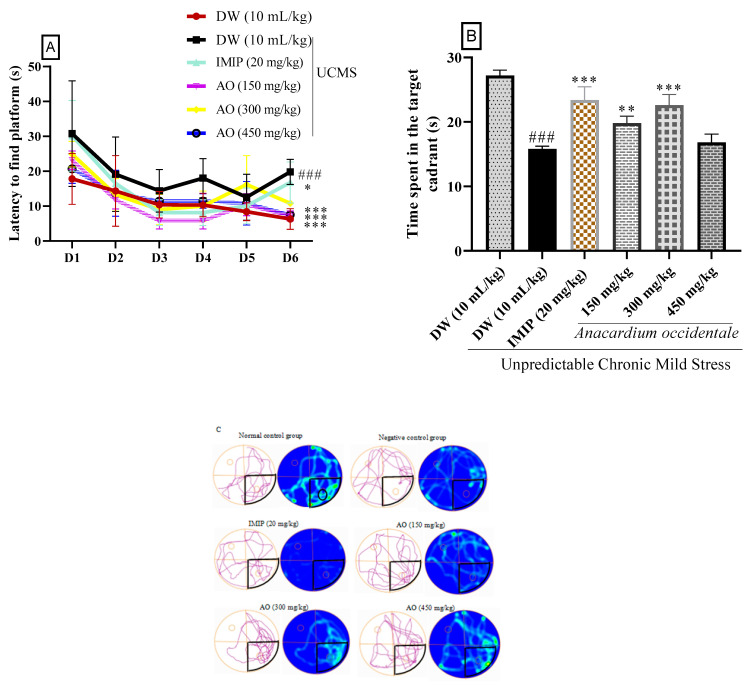
Effect of *A. occidentale* fruit kernel extract on escape latency (**A**) and target quadrant dwelling time (**B**), path length, and heat plots (**C**) in the Morris water maze. Each bar indicates the average ± SD; n = 5; IMIP = imipramine; AO = *Anacardium occidentale*; UCMS = unpredictable chronic mild stress; ^###^
*p* < 0.001 significant difference compared to the unstressed group; * *p* < 0.05, ** *p* < 0.01, *** *p* < 0.001 significant difference compared to the UCMS control group; 
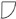
 = platform location quadrant.

**Figure 9 brainsci-13-01561-f009:**
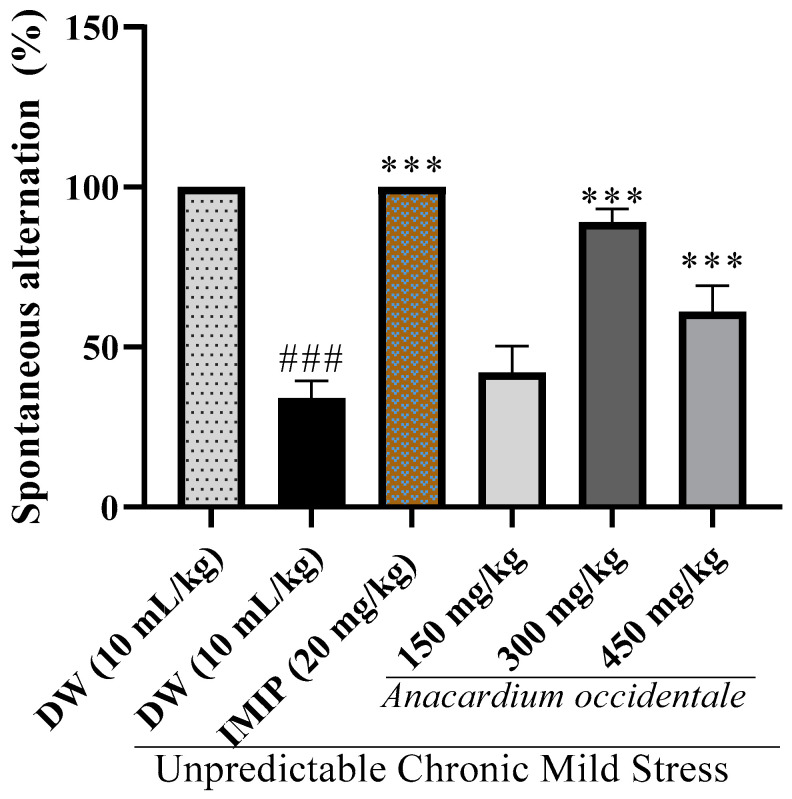
Effect of *A. occidentale* fruit kernel extract on spontaneous alternation in the Y-maze test. Each bar indicates the mean ± SD; n = 5; DW = distilled water; IMIP = imipramine; ^###^
*p*< 0.01 significant difference compared with the unstressed control; *** *p* < 0.001 significant difference compared with the UCMS control group.

**Figure 10 brainsci-13-01561-f010:**
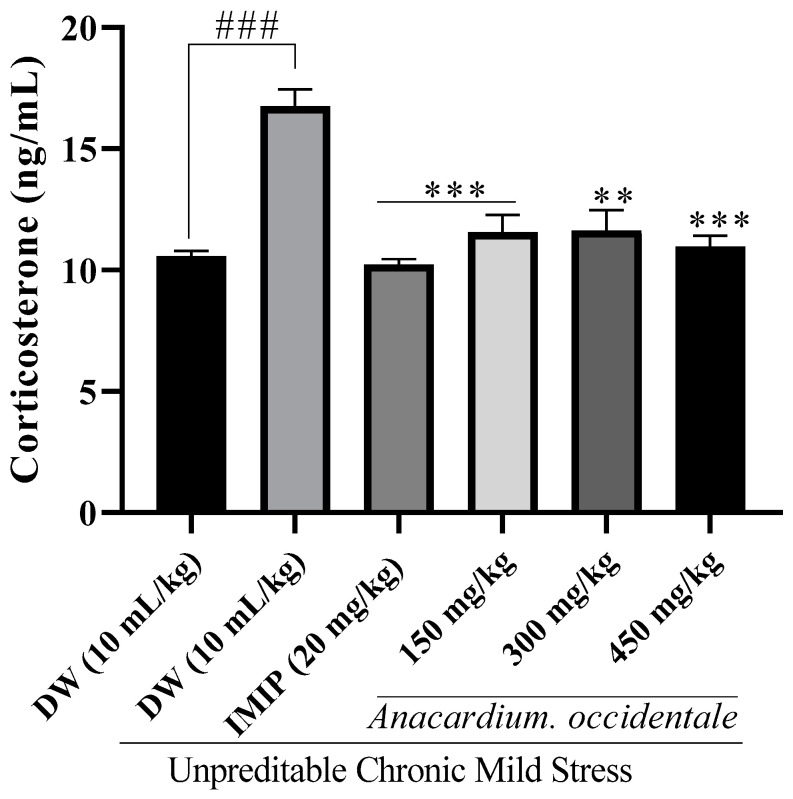
Impact of *A. occidentale* fruit kernel hydroethanolic extract on corticosterone concentration. n = 3; each result represents means ± SD; ^###^ *p*< 0.001 compared with the unstressed group; ** *p* < 0.01, *** *p* < 0.001 compared with the UCMS control group.

**Figure 11 brainsci-13-01561-f011:**
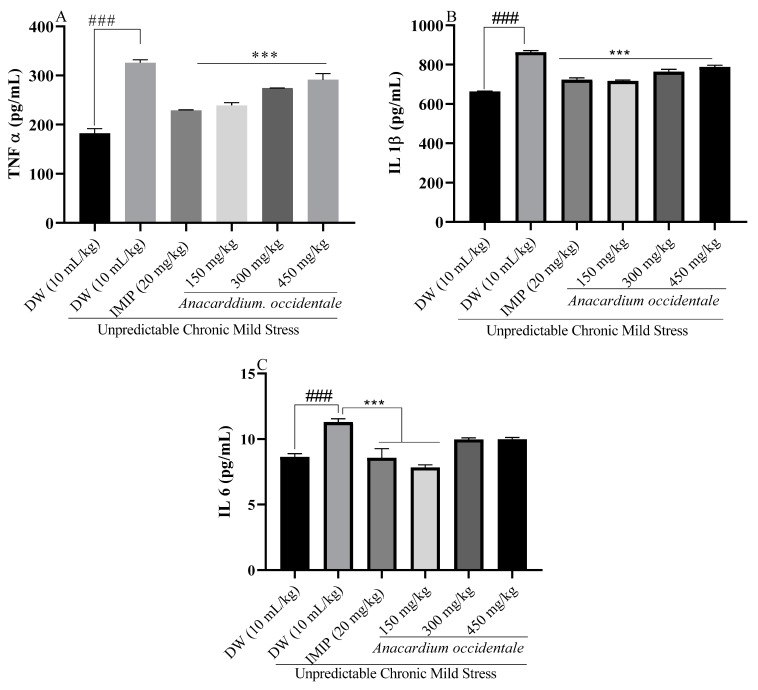
Effect of the hydroethanolic extract of *A. occidentale* fruit kernel on tumor necrosis factor (**A**), interleukin 1β (**B**), and 6 (**C**) expression. n = 3; each result represents means ± SD; ^###^
*p* < 0.001 compared with unstressed group; *** *p* < 0.001 compared with the UCMS control group.

**Figure 12 brainsci-13-01561-f012:**
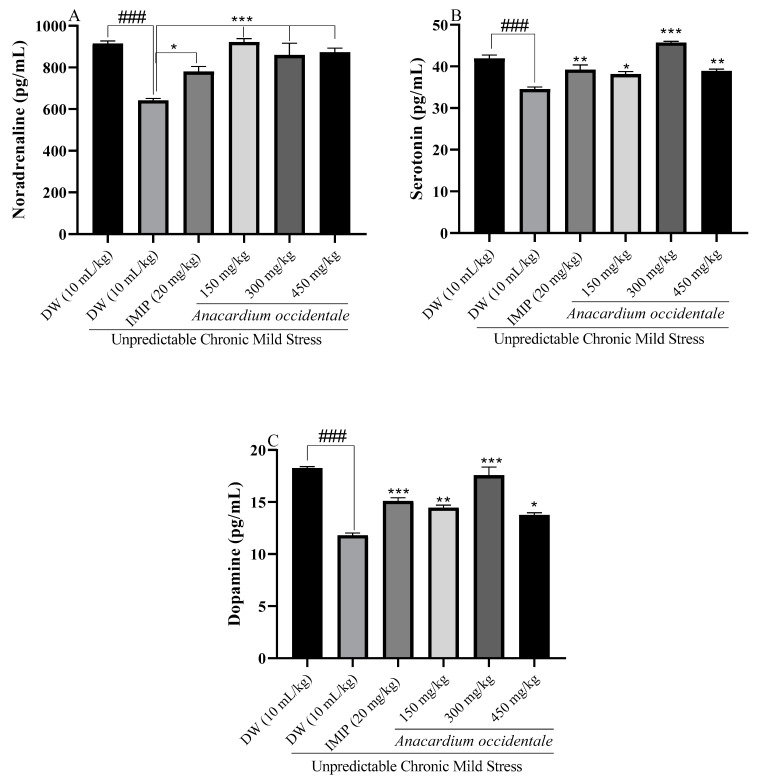
Impact of the hydroethanolic extract of *A. occidentale* fruit kernel on noradrenaline (**A**), serotonin (**B**), and dopamine (**C**) expression. n = 3; each result represents means ± SD; ^###^ *p* < 0.001 compared with unstressed group; * *p* < 0.05, ** *p* < 0.01, *** *p* < 0.001 compared to the UCMS control group.

**Figure 13 brainsci-13-01561-f013:**
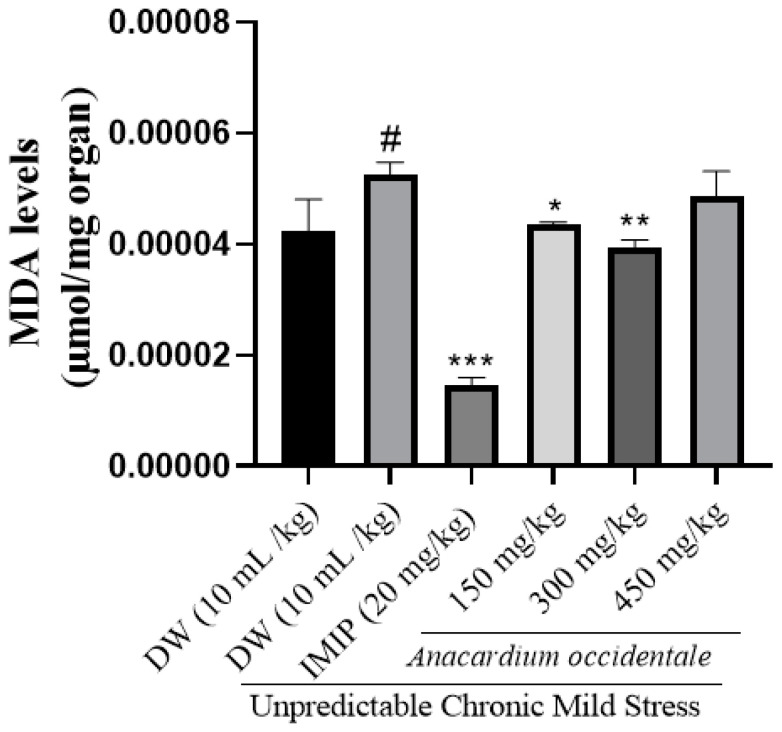
Effect of hydroethanolic extract of *A. occidentale* fruit kernel on MDA concentration n = 3; each result represents mean ± SD; ^#^
*p <* 0.05, compared to the unstressed group; * *p* < 0.05, ** *p* < 0.01, *** *p* < 0.001 compared to the UCMS control group.

**Figure 14 brainsci-13-01561-f014:**
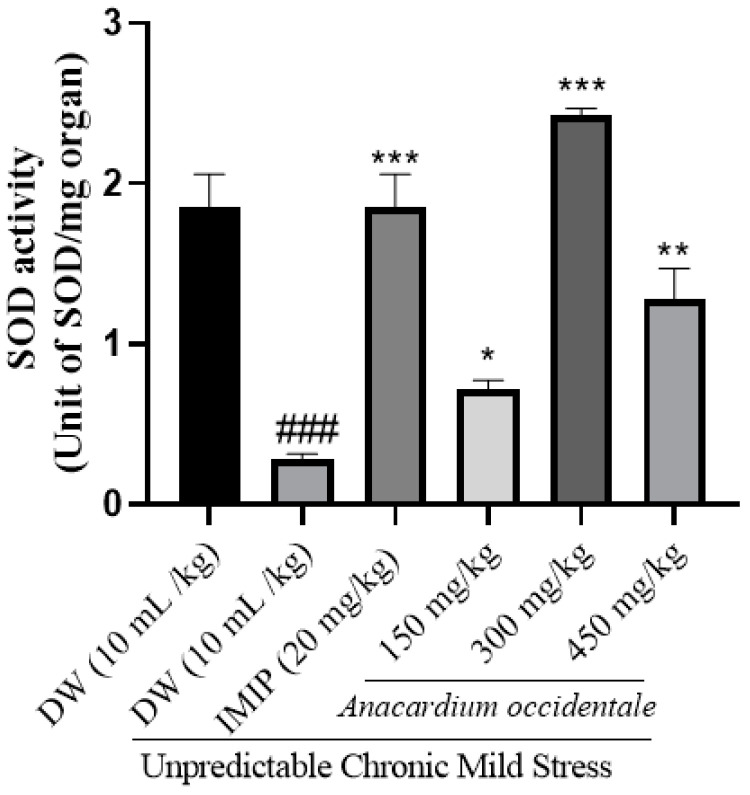
Impact of *A. occidentale* fruit kernel hydroethanolic extract on SOD activity; n = 3; each result represents means ± SD; ^###^ *p* < 0.001 compared to the unstressed group; * *p* < 0.05, ** *p* < 0.01, *** *p* < 0.001 compared to the UCMS control group.

**Figure 15 brainsci-13-01561-f015:**
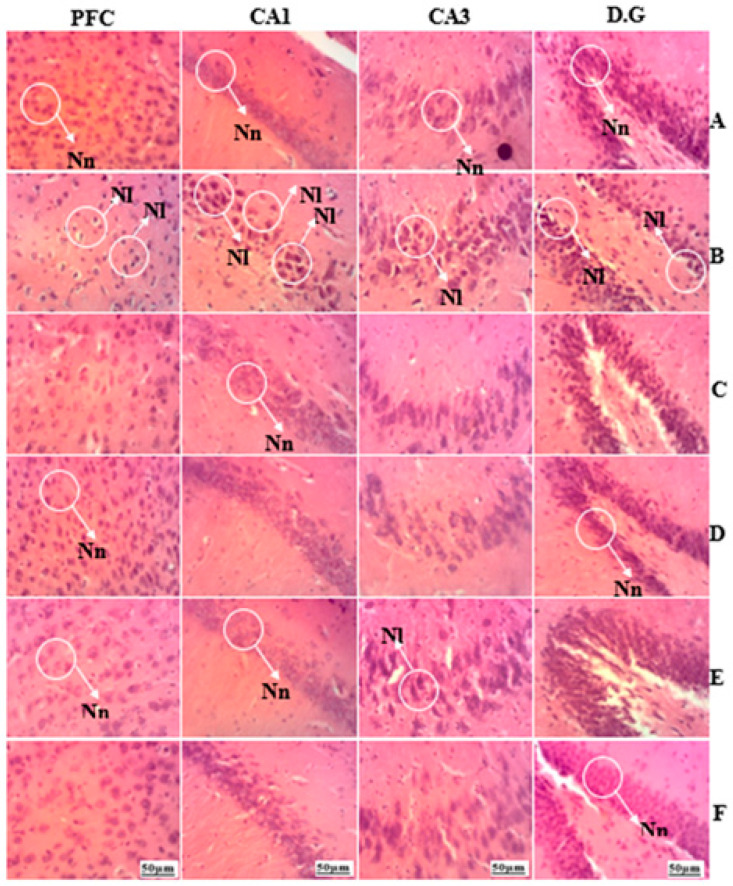
Effect of the hydroethanolic extract of *A. occidentale* fruit kernel on hippocampus and prefrontal cortex neurons. PFC: prefrontal cortex; CA1 and CA3; D.G: dentate gyrus; (**A**): normal group; (**B**): negative group; (**C**): positive group, (**D**): *Anacardium occidentale* (150 mg/kg); (**E**): *Anacardium occidentale* (300 mg/kg); (**F**): *Anacardium occidentale* (450 mg/kg); Nn = Normal neuron; Nl = Neuronal loss.

**Figure 16 brainsci-13-01561-f016:**
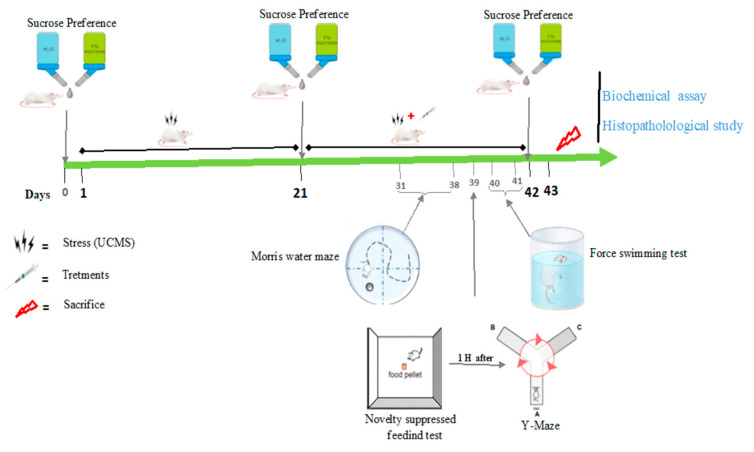
Experimental design of the study.

**Table 1 brainsci-13-01561-t001:** Names, structures, and formulae of the identified compounds from the *A. occidentale* fruit kernel.

N°	Compounds	Structures	Formulas
1	Limonene	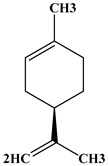	C_10_H_16_
2	Â-sitosterol	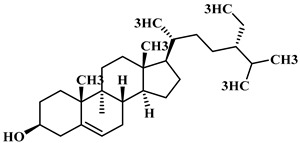	C_29_H_50_O
3	Hydroxyhopanon	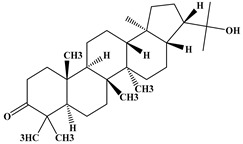	C_30_H_50_O_2_
4	Kaempferol	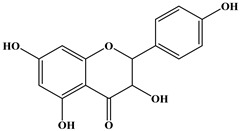	C_15_H_10_O_6_
5	Catechin	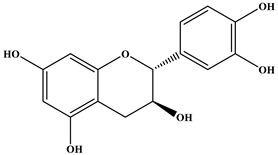	C_15_H_14_O_6_
6	Lupane	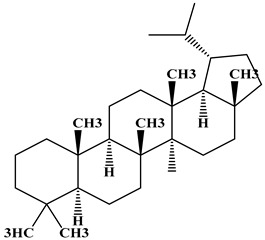	C_30_H_52_
7	Quercetin	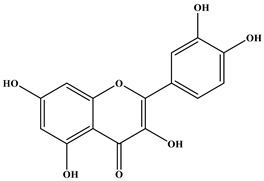	C_15_H_10_O_7_
8	Isoquercetin	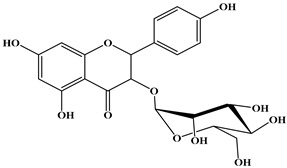	C_21_H_20_O_12_
9	Hispiduline	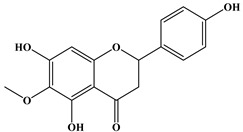	C_16_H_12_O_6_
10	Ursolic acid	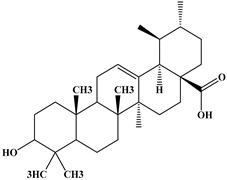	C_30_H_48_O_3_
11	Oleanolic acid	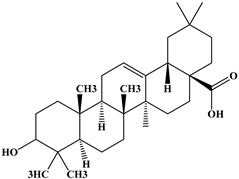	C_30_H_48_O_3_

**Table 2 brainsci-13-01561-t002:** % (IC 50%).

Sample	Inhibition Percentage (%)	IC_50_ (ìg/mL)
Anacardium occidentale Extract	69.32	150
BHT	74.05	100

AO = Anacardium occidentale; BHT = butyl hydroxytoluene; IC 50 = inhibitory concentration 50%.

**Table 3 brainsci-13-01561-t003:** Chronology of exposure to stressors and their duration.

Days	Stressors	Duration
1–2	Isolation	48 h
3	Food deprivation	24 h
4	Sound stimulation + Night illumination	1 h +12 h
5	Water deprivation	24 h
6	Wet litter (250 mL/100 g)	24 h
7	Forced swimming 30 °C	10 min
8	Physical restraint	2 h
9	Sound stimulation + Night illumination	1 h +12 h
10	Wet litter (250 mL/100 g)	24 min
11	Forced swimming at 30 °C	10 min
12–13	Isolation	48 h
14	Water deprivation	24 h
15	Food deprivation	24 h
16	Physical restraint	2 h
17	Wet litter (250 mL/100 g)	24 h
18	No stress	24 h
19	Sound stimulation + Night illumination	1 h + 12 h
20	Forced swimming 30 °C	5 min
21	Food deprivation	24 h
22	Wet litter (250 mL/100 g)	24 h
23	Sound stimulation + Night illumination	1 h + 12 h
24	Food deprivation	24 h
25	Wet litter (250 mL/100 g)	24 h
26–27	Isolation	48 h
28	Physical restraint	2 h
29	Water deprivation	24 h
30	No stress	24 h
31	Forced swimming 30 °C	5 min
32–33	Isolation	48 h
34	Water deprivation	24 h
35	Wet litter (250 mL/100 g)	24 h
36	No stress	24 h
37	Sound stimulation + Night illumination	1 h + 12 h
38	Food deprivation	24 h
39	Physical restraint	2 h
40	Wet litter (250 mL/100 g)	24 h
41	Sound stimulation + Night illumination	1 h + 12 h
42	No stress	24 h

h = hour; min = minute.

## Data Availability

All data used are available on request.

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
