# Peer review of "Cashew (Anacardium occidentale) Extract: Possible Effects on Hypothalamic–Pituitary–Adrenal (HPA) Axis in Modulating Chronic Stress"

_brainsci, 2023, doi:10.3390/brainsci13111561_

Round 1

Reviewer 1 Report

Comments and Suggestions for Authors

This research shows that hydroethanolic extract of Anacardium occidentale (AO) induce antidepressant-like effects in unpredictable chronic mild stress (UMCS) in rats with possible involvement of the monoaminergic neurotransmission. The results can be of relevance for the study and identification of potential ne antidepressant drugs and their mechanism of action. Overall this a well-designed study and provides very interesting results. With some minor editing to the manuscript this paper is deserving of publication.

Some issues which raised my concerns, I have listed below:

-          Do you have any suggestions/possible explanations for the lack of effect of a dose 300 mg/kg on sucrose consumption in anhedonic rats (Fig. 4C)?

-          Could you more deeply discuss the results of FST, please? The Authors show the results based on the three parameters and it is worth to discuss the differences between them.

Corrections:

Abstract: please add the abbreviation for Anacardium occidentale.

Keyword: amnesia is not a suitable keywords for the manuscript. I suggest replacing this word the “stress”.

Page 2 (introduction):

 – please explain the abbreviation HPA

- “imipraminics” I suppose that it is a mistake

- Please correct the reference 19, check the surname of the first author

- Page 11, line 4: In addition, a significant increase (decrease?) of latency was observed after the administration of the extract whatever the dose used, compared with the negative control group.

Author Response

Dear reviewer, first we would like to thank you for your positive and encouraging comments regarding our research paper. Each of your insights have served to strengthen our manuscript and we have made changes to reflect them.

This research shows that hydroethanolic extract of Anacardium occidentale (AO) induce

antidepressant-like effects in unpredictable chronic mild stress (UMCS) in rats with possible

involvement of the monoaminergic neurotransmission. The results can be of relevance for the study and identification of potential ne antidepressant drugs and their mechanism of action. Overall this a well-designed study and provides very interesting results. With some minor editing to the manuscript this paper is deserving of publication.

Major points:

Do you have any suggestions/possible explanations for the lack of effect of a dose 300 mg/kg on sucrose consumption in anhedonic rats (Fig. 4C).

Some substances, particularly natural compounds, can exhibit biphasic Dose-Response, where lower and higher doses produce different effects [1]. The 300 mg/kg dose might be in a range where it neither enhances nor inhibits sucrose consumption significantly. The dose of 300 mg/kg of the hydroethanolic extract of Anacardium occidentale had a different effect on sucrose consumption compared to other doses (150 mg/kg and 450 mg/kg). Specifically, we observed that this particular dose falls within a "neutral" zone where it neither enhances nor inhibits sucrose consumption significantly in anhedonic rats. So, in summary, the lack of a significant effect of the 300 mg/kg dose on sucrose consumption in anhedonic rats may be due to the biphasic properties of the secondary metabolic of the Anacardium occidentale extract. At this specific dose, the extract might not have triggered the desired response in these animals, but lower or higher doses could potentially lead to different and perhaps more significant effects.

By searching in the literature we found that most of the major secondary metabolites in our extracts exhibit a biphasic dose-response effect [1-5].

[1] Jodynis-Liebert, J.; & Kujawska, M. Biphasic dose-response induced by  phytochemicals: experimental evidence. Journal of Clinical Medicine 2020, 9(3), 718.

[2] Oh, S. M.; Kim, Y. P.; & Chung, K. H. Biphasic effects of kaempferol on the estrogenicity in human breast cancer cells. Archives of pharmacal research 2006, 29, 354-362.

[3] Valsa, A. K.; Ushaumari, B.; & Vijayalakshmi, N. P. Effect of catechin on lipid metabolism. Journal of clinical biochemistry and nutrition 1995, 19(3), 175-182.

[4] Wang, H.; Sim, M. K.; Loke, W. K.; Chinnathambi, A.; Alharbi, S. A.; Tang, F. R.; & Sethi, G. Potential protective effects of ursolic acid against gamma irradiation-induced damage are mediated through the modulation of diverse inflammatory mediators. Frontiers in Pharmacology 2017, 8, 352.

[5] Vargas, A. J.; & Burd, R. Hormesis and synergy: pathways and mechanisms of quercetin in cancer prevention and management. Nutrition reviews 2010, 68(7), 418-428.

Could you more deeply discuss the results of FST, please? The Authors show the results based on the three parameters and it is worth to discuss the differences between them

The Forced Swimming Test (FST) results (Figure 6), provide valuable insights into the impact of the hydroethanolic extract of Anacardium occidentale (AO) on the behavior of rats subjected to chronic mild unpredictable stress (UCMS). The FST is a commonly used behavioral test for assessing depressive-like behaviors and the potential antidepressant effects of substances.

Swimming time is a parameter that measures the time that rats spent actively swimming and it’s considered an active escape behavior in the FST or a measure of struggling behavior. In our study, UCMS exposure led to a significant reduction in swimming behavior compared to non-exposed rats, which indicate a depressive-like state (p < 0.001) (figure 6A and B). This result aligns with the expectation that animals subjected to chronic stress are more likely to display passive and despairing behaviors. However, treatment of anhedonic rats with the hydroethanolic extract of AO, as well as imipramine, increased swimming time. We observe that the lower dose (150 mg/kg) led to a more pronounced raise in swimming time compared to the 300 and 400 mg/kg doses. This effect may be linked to the biphasic dose-response characteristics of the hydroethanolic extract of Anacardium occidentale (AO) [40-44]. Our results suggest that both AO and imipramine have an antidepressant-like effect by promoting active coping strategies in response to the stressor.

Climbing time in the FST is another active response, representing the duration rats spent trying to climb out of the water, and reflecting the animal's efforts to escape the stressful condition. The findings demonstrate that rats exposed to UCMS had shorter climbing time compared to non-exposed rats, indicating a significantly diminished capacity to engage in active escape attempts (p < 0.01) (figure 6A and B). Treatment of anhedonic rats with AO extract and imipramine, led to a significant rise in climbing behavior (p < 0.001). Similar to time to swim, both AO and imipramine promote active and goal-directed behaviors in response to the stressor, indicating a reduction in depressive-like behaviors. 

While immobility in the FST, which reflects a passive and despairing state. UCMS exposure augmented significantly the immobility time (p < 0.001) (figure 6C) when compared with non-exposed rats, is generally associated with depressive-like behavior. However, all doses 150 mg/kg, 300 mg/kg, and 450 mg/kg of the AO extract, along with imipramine (20 mg/kg), demonstrated a significant reduction (p < 0.001) in immobility time when contrasted with the UCMS control group. The reduction in immobility time implies an antidepressant-like effect, as the rats become less passive and more involved in active behaviors. In addition, a dose-response effect was observed, with a higher decrease recorded for the lower dose (15mg/mg) followed by the doses 300 mg/kg and 400 mg/kg. Similar to swimming time results, this pattern can be attributed to the biphasic dose-response properties of the hydroethanolic extract of Anacardium occidentale (AO) [40-44].

Taken together, the results from the FST indicate that UCMS exposure induces depressive-like behaviors, characterized by increased immobility and reduced swimming and climbing. The treatment with the AO extract at all doses and the reference drug imipramine led to significant improvements in the animals' behavior. These improvements include increased active escape behaviors (swimming and climbing) and reduced passive despair (immobility).

These findings suggest that the AO hydroethanolic extract possesses antidepressant-like effects, as it was able to reverse the depressive behaviors induced by chronic stress. The FST results align with the earlier findings of the sucrose preference test and further support the potential antidepressant effects of this extract.

[40] Jodynis-Liebert, J.; & Kujawska, M. Biphasic dose-response induced by phytochemicals: experimental evidence. Journal of Clinical Medicine 2020, 9(3), 718.

[41] Oh, S. M.; Kim, Y. P.; & Chung, K. H. Biphasic effects of kaempferol on the estrogenicity in human breast cancer cells. Archives of pharmacal research 2006, 29, 354-362.

[42] Valsa, A. K.; Ushaumari, B.; & Vijayalakshmi, N. P. Effect of catechin on lipid metabolism. Journal of clinical biochemistry and nutrition 1995, 19(3), 175-182.

[43] Wang, H.; Sim, M. K.; Loke, W. K.; Chinnathambi, A.; Alharbi, S. A.; Tang, F. R.; & Sethi, G. Potential protective effects of ursolic acid against gamma irradiation-induced damage are mediated through the modulation of diverse inflammatory mediators. Frontiers in Pharmacology 2017, 8, 352.

[44] Vargas, A. J.; & Burd, R. Hormesis and synergy: pathways and mechanisms of quercetin in cancer prevention and management. Nutrition reviews 2010, 68(7), 418-428.

Minor points

Abstract: please add the abbreviation for Anacardium occidentale.

Thank you very much for your comments. We added the requested abbreviation for Anacardium occidentale “AO” .

Keyword: amnesia is not a suitable keywords for the manuscript. I suggest replacing this word the “stress”.

Thank you very much for your comments. We changed the word amnesia to stress

Page 2 (introduction):

Please explain the abbreviation HPA.

Thank you very much for your comments. We added the explanation of the abbreviation to HPA “Hypothalamic-Pituitary-Adrenal”

“imipraminics” I suppose that it is a mistake

Thank you very much for your comments. We corrected the word in the text « imipramine »

Please correct the reference 19, check the surname of the first author

Thank you very much for your comments. We corrected the surname of the first author in the text: « Dharamveer »

Page 11, line 4: In addition, a significant increase (decrease?) of latency was observed after
the administration of the extract whatever the dose used, compared with the negative control group.

Thank you. It is “decrease”. We have changed now it in the text. Thank you again very much for your kindness.

Reviewer 2 Report

Comments and Suggestions for Authors

 Authors have investigated the role of Anacardium occidentale in the UCMS rat model of depressive-like behaviors. Several behavioral studies related to memory and depression were conducted. Biochemical estimations for oxidative stress and neuroinflammation were performed. Histopathology of brain regions was also examined. The study is well-conduced; however, I have several serious issues with the study parameters and other minor limitations.

1. There are so many behavioral studies, specifically, FST and MWM test, that might inflict extra stress on the animals. Won’t it affect the already established stress model? Moreover, MWM is mainly for the assessment of spatial memory. How this UCMS model would affect this specific type of memory? The same goes for the Y-maze test, which is generally done for memory assessment. Furthermore, forced swimming itself was one of the stress inducers in the UCMS model. 8 days for MWM is also too long. It could have been done in 5 days.

2. The abstract mentioned MWM results, but not the main behavioral results of depression-related parameters.

3. Were they cashew nuts or almonds? Please clarify, as both things have been mentioned in the manuscript.

4. In the Introduction section, the UCMS model can be introduced briefly, as to how it is related to depression.

5. Authors mentioned the prevalence in some of the continents. Asia could also be included.

6. The authors didn’t measure the initial body weight from day 1 to find out the effect of initial stress on body weight.

7. What could be the rationale behind more depressive behavior with higher doses in FST? Same with lipid peroxidation.

8. “The decrease in cytokines could confer central anti-inflammatory effect to the extract, resulting in better bioavailability of monoamines and neuronal cells survival”. Through which mechanisms decreased cytokines will lead to improved monoamines?

9. Haematoxylin/Eosin or nigrosine/eosin? Which one should have been taken? Which one is suitable?

10. For a research article, number of references seems larger.

11. First line of the Introduction, please correct the grammar “one of most major”. The same goes for other grammatical errors throughout the manuscript. It must be improved with the help of a native English speaker or suitable online tools. Furthermore, proper upper- and lower-case use, punctuation, etc. should be taken care of.

12. Isn’t the number of samples n=3 too low?

13. The abbreviations AO and even others should be expanded and used in brackets when used for the first time.

Minor comments

14. Instead of hypotensive, mention anti-hypertensive in the Introduction section.

15. In vitro and in vivo can be written in italics throughout the manuscript.

16. Fig 4c, its chronique or chronic?

17. Is it ELIZA or ELISA?

18. Alpha can be put as α

Comments on the Quality of English Language

English grammar needs extensive improvement through proofreading.

Author Response

Reviewer 2

Dear Reviewer, Thank you very much for your valuable comments and suggestions and for taking the time to point out options to greatly improve the quality of our manuscript. We agree with all your comments, and we corrected the manuscript point by point accordingly.

Authors have investigated the role of Anacardium occidentale in the UCMS rat model of depressive-like behaviors. Several behavioral studies related to memory and depression were conducted. Biochemical estimations for oxidative stress and neuroinflammation were performed. Histopathology of brain regions was also examined. The study is well conducted; however, I have several serious issues with the study parameters and other minor limitations.

There are so many behavioral studies, specifically, FST and MWM test, that might inflict extra stress on the animals. A) Won’t it affect the already established stress model? Moreover, MWM is mainly for the assessment of spatial memory. B) How this UCMS model would affect this specific type of memory? The same goes for the Y-maze test, which is generally done for memory assessment. Furthermore, forced swimming itself was one of the stress inducers in the UCMS model. C) 8 days for MWM is also too long. It could have been done in 5 days.

  1. Won’t it affect the already established stress model?

Thank you for your comments. The addition of the forced swim test (FST) and the Morris water maze (MWM) into our study was a carefully considered decision to improve the comprehensiveness of our research. Our established stress model plays a central role in our research, as it serves as a reference for evaluating the effects of Anacardiumouest (AO) hydroethanolic extract. By subjecting animals to this standardized stress model, we ensured a consistent and controlled environment for inducing stress. The inclusion of these additional behavioral tests into our study provides a more comprehensive assessment of the effects of AO extract. This allows us to explore its potential to alleviate depressive behaviors and stress-induced cognitive deficits. We understand that these tests can cause additional stress to animals, and it is essential to recognize this. However, our main goal is to determine whether the extract can counteract the harmful effects of stress. It is essential to emphasize that we have been careful in the selection of our animals. We chose them based on their anhedonic reactions to UCMS, making sure they were actually exhibiting stress-related behaviors before conducting the experiments.

  1. Moreover, MWM is mainly for the assessment of spatial memory. How this UCMS model would affect this specific type of memory? The same goes for the Y-maze test, which is generally done for memory assessment. Furthermore, forced swimming itself was one of the stress inducers in the UCMS model.

The reviewer raised a valid observation concerning the potential impact of the Chronic Unpredictable Mild Stress (UCMS) model on the assessment of spatial memory using the Morris Water Maze (MWM) test.  Although the UCMS model mainly induces depressive behavior and stress-related changes, it can indirectly influence various cognitive functions, namely spatial memory [1]. Chronic stress has been associated with alterations in cognitive processes, and the UCMS model is likely to contribute to the overall cognitive context in which spatial memory is assessed [1]. In addition, the MWM test may also be sensitive to mood-related factors, and the combination of the UCMS model and the MWM test provides an opportunity to explore potential links between stress-induced mood alterations and cognitive performance [2].

The same applies to the Y maze test. Potential adverse effects of UCMS include impaired working memory, reduced spatial recognition, altered exploratory behavior and disrupted contextual memory [3-4].

 [1] Song, L; Che, W.; Min-Wei, W.; Murakami, Y.; & Matsumoto, K. Impairment of the spatial learning and memory induced by learned helplessness and chronic mild stress. Pharmacology Biochemistry and Behavior 200683(2), 186-193.

[2] Liu, S. C.; Hu, W. Y.; Zhang, W. Y.; Yang, L.; Li, Y.; Xiao, Z. C.; ... & He, Z. Y. Paeoniflorin attenuates impairment of spatial learning and hippocampal long-term potentiation in mice subjected to chronic unpredictable mild stress. Psychopharmacology 2019236, 2823-2834.

[3] Van Boxelaere, M.; Clements, J.; Callaerts, P.; D’Hooge, R.; & Callaerts-Vegh, Z. Unpredictable chronic mild stress differentially impairs social and contextual discrimination learning in two inbred mouse strains. PLoS One 2017, 12(11), e0188537.

[4] Mineur, Y. S.; Belzung, C.; & Crusio, W. E. Effects of unpredictable chronic mild stress on anxiety and depression-like behavior in mice. Behavioural brain research 2006, 175(1), 43-50.

  1. 8 days for MWM is also too long. It could have been done in 5 days.

Thank you for this valuable input. In our investigation study, we selected an 8-day duration for the Morris Water Maze (MWM) testing to ensure a detailed and comprehensive assessment of spatial memory. Our choice is in line with the protocol described by [3-4], where rats were trained for 8 days in the MWM. However, we value the reviewer's suggestion and will certainly consider it for potential modifications in future research designs.

 [3] Jayalakshmi, K.; Singh, S. B.; Kalpana, B.; Sairam, M.; Muthuraju, S.; & Ilavazhagan, G. N-acetyl cysteine supplementation prevents impairment of spatial working memory functions in rats following exposure to hypobaric hypoxia. Physiology & behavior 2007, 92(4), 643-650.

[4] Derecki, N. C.; Quinnies, K. M.; & Kipnis, J. Alternatively activated myeloid (M2) cells enhance cognitive function in immune compromised mice. Brain, behavior, and immunity 2011, 25(3), 379-385.

The abstract mentioned MWM results, but not the main behavioral results of depression-related parameters

The main behavioral results of depression-related parameters has been added to the abstract

Abstract:. Depression presents a significant global health burden, necessitating the search for ef-fective and safe treatments. The investigation aims to assess the antidepressant effect of the hy-droethanolic extract of Anacardium occidentale (AO) on depression-related behaviors in rats. The depression model involved 42 days Unpredictable Chronic Mild Stress (UCMS) exposure and assessed using the sucrose preference and the forced swimming (FST) test. Additionally, memory-related aspects were examined using the tests Y-maze and Morris water maze (MWM), following 21 days of treatment with varying doses of the AO extract (150, 300, and 450 mg/kg) and Imipramine (20 mg/kg), commencing on day 21. The monoamines (norepinephrine, seroto-nin and dopamine), oxidative stress markers (MDA and SOD) and cytokines levels (IL-1β,IL-6 and TNF-?) within the brain were evaluated. Additionally, the concentration of blood corti-costerone was measured. Treatment with A. occidentale significantly alleviated UCMS-induced and depressive-like behaviors in rats. This was evidenced by the ability of the extract to prevent further decreases in body mass, increase sucrose consumption, reduce immobility time in the test Forced Swimming, improve cognitive performance in both tests Y-Maze and the Morris Wa-ter Maze by increasing the target quadrant dwelling time and spontaneous alternation percent-age, and promote faster feeding behavior in the Novelty-Suppressed Feeding Test. It also de-creased pro-inflammatory cytokines, corticosterone and MDA levels and increased the monoam-ines levels and SOD activity.  HPLC-MS analysis revealed the presence of triterpenoids com-pounds (ursolic acid, oleanolic acid, lupane) and polyphenols (catechin quercetin and kaempferol). These results evidenced the antidepressant effects of the AO which might involve the corticosterone and monoaminergic regulation, antioxidant and anti-inflammatory activities.

Were they cashew nuts or almonds? Please clarify, as both things have been mentioned in the manuscript.

Thank you for your comments. It was the cashew nuts; we have corrected it in the manuscript.

In the Introduction section, the UCMS model can be introduced briefly, as to how it is related to depression.

The UCMS model is a widely recognized preclinical research paradigm used to mimic depressive and anxiety-like behaviors in rodents through chronic exposure to mild psychosocial stressors [18]. This model has demonstrated face, predictive, and construct validity, making it one of the few models where chronic, rather than acute, monoaminergic antidepressant administration proves effective [19]. The UCMS protocol induces a depressive-like state in animals, akin to human depression characterized by apathy and anhedonia [19]. It evaluates stress responses and antidepressant effects through behavioral measures such as spontaneous motivation, spontaneous grooming behavior, and appetence for pleasurable food [19]. This model's high degree of unpredictability and uncontrollability of stressors, along with its use of mild stressors, enhances its relevance to human conditions [18]. Importantly, the UCMS model offers a translational bridge for investigating the pathophysiology of depression and testing potential therapeutic pharmacological agents in rodents [18].

[18] Frisbee, J.C.; Brooks, S.D.; Stanley, S.C.; d'Audiffret, A.C. An Unpredictable Chronic Mild Stress Protocol for Instigating Depressive Symptoms, Behavioral Changes and Negative Health Outcomes in Rodents. J. Vis. Exp 2015. (106), e53109, doi:10.3791/53109.

[19] Nollet, M. Models of depression: Unpredictable chronic mild stress in mice. Current Protocols 2021, 1, e208. doi: 10.1002/cpz1.208.

Authors mentioned the prevalence in some of the continents. Asia could also be included.

Thank you very much for your comments. It has been added in the introduction part:

Its prevalence varied across regions, with rates of 16 % in the Eastern Mediterranean Region, 9 % in the African Region, 12 % in the European Region, 15 % in the Region of the Americas, 21 % in the Western Pacific Region, and 27 % in the South East Asia Region [4].

The authors didn’t measure the initial body weight from day 1 to find out the effect of initial stress on body weight.

After these 21 days of the Unpredictable chronic mild stress, the sucrose preference test and the novelty suppressed feeding test were performed to assess the stress development in animals. However, at the start of the UCMS protocol, the initial weight of all animals was recorded. These data are available and can be integrated into the Figure 3 if necessary.

What could be the rationale behind more depressive behavior with higher doses in FST? Same with lipid peroxidation.

In the literature, some substances, particularly natural compounds, showed biphasic Dose-Response, where lower and higher doses produce different effects [1]. The lack of a significant effect with a higher dose observed in both FST and lipid peroxidation could be attributed to the biphasic properties of the secondary metabolic of the Anacardium occidentale extract. 

By searching in the literature, we found that most of the major secondary metabolites in our extracts exhibit a biphasic dose-response effect [1-5].

[1] Jodynis-Liebert, J.; & Kujawska, M. Biphasic dose-response induced by phytochemicals: experimental evidence. Journal of Clinical Medicine 2020, 9(3), 718.

[2] Oh, S. M., Kim, Y. P., & Chung, K. H. Biphasic effects of kaempferol on the estrogenicity in human breast cancer cells. Archives of pharmacal research 2006, 29, 354-362.

[3] Valsa, A. K.; Ushaumari, B.; & Vijayalakshmi, N. P. Effect of catechin on lipid metabolism. Journal of clinical biochemistry and nutrition 1995, 19(3), 175-182.

[4] Wang, H.; Sim, M. K.; Loke, W. K.; Chinnathambi, A.; Alharbi, S. A.; Tang, F. R.; & Sethi, G. Potential protective effects of ursolic acid against gamma irradiation-induced damage are mediated through the modulation of diverse inflammatory mediators. Frontiers in Pharmacology 2017, 8, 352.

[5] Vargas, A. J.; & Burd, R. Hormesis and synergy: pathways and mechanisms of quercetin in cancer prevention and management. Nutrition reviews 2010, 68(7), 418-428.

“The decrease in cytokines could confer central anti-inflammatory effect to the extract, resulting in better bioavailability of monoamines and neuronal cells survival”. Through which mechanisms decreased cytokines will lead to improved monoamines?

The decrease in cytokine levels in response to the AO extract observed in our finding, can create a more favorable environment for the reuptake, synthesis, and release of monoamines in the central nervous system via several mechanisms. Firstly, the anti-inflammatory properties of AO extract reduce the inflammatory response, by directly inhibiting the release or activity of pro-inflammatory cytokines including tumor necrosis factor alpha (TNF-α) and interleukins (IL-1β and IL-6). By reducing the levels of these cytokines, the extract helps to attenuate the overall inflammatory response. This anti-inflammatory effect increases the availability of the monoamines identified in our study, such as serotonin, noradrenaline and dopamine, which play a crucial role in mood regulation.. This anti-inflammatory effect enhances the availability of monoamines identified in our study, like serotonin, norepinephrine, and dopamine, which play crucial roles in mood regulation. Secondly, by lowering the levels of pro-inflammatory cytokines, the extract mitigates the impact of stress on the body, as chronic stress contributes to elevated cytokine levels. Lower stress levels create a less hostile environment for the central nervous system, promoting the normal functioning of monoaminergic systems. Finally, the extract's antioxidant activity protects monoamines from oxidative damage caused by chronic stress and inflammation, thereby promoting their stability and functionality. In conclusion, reduced inflammation, stress, and oxidative stress all collectively contribute to more balanced monoamines in the central nervous system, potentially relieving depressive symptoms and supporting healthier mood regulation.

Haematoxylin/Eosin or nigrosine/eosin? Which one should have been taken? Which one is suitable?

Thank you for your comments. We corrected in the text this omission, Hippocampal and prefrontal cortex regions were stained with Haematoxylin/eosin. We have added it to the manuscript

For a research article, number of references seems larger.

Thank you for your comment. We would like to address this concern by providing the following explanations:

The extensive reference list is a reflection of our numerous results, enabling us to conduct a thorough analysis of our findings in the context of the pertinent literature.

First line of the Introduction, please correct the grammar “one of most major”. The same goes for other grammatical errors throughout the manuscript. It must be improved with the help of a
native English speaker or suitable online tools. Furthermore, proper upper- and lower-case use, punctuation, etc. should be taken care of.

Thank you very much for your comments. We have made the requested grammar change

Isn’t the number of samples n=3 too low

In carrying out the behavioral tests, each group or treatment consisted of 5 animals. However, in biochemical assays 3 animals per group were considered. Biochemical tests may not require as large a sample size, as they are often more precise and less variable, as well as for ethical reasons.

The abbreviations AO and even others should be expanded and used in brackets when used for the first time.

Thank you very much for your comments. We have made the requested change

Minor comments

Instead of hypotensive, mention anti-hypertensive in the Introduction section.

Thank you very much for your comments. We have replaced hypotensive with anti-hypertensive in the manuscript

In vitro and in vivo can be written in italics throughout the manuscript.

Thank you very much for your comments. We have made the requested change in the manuscript

Fig 4c, its chronique or chronic?

Thank you very much for your comments. It’s Chronic, We have changed it the manuscript

Is it ELIZA or ELISA?

Thank you very much for your comments. It’s ELISA, We have changed it the manuscript

Alpha can be put as α

Thank you very much for your comments. We have made the requested change in the manuscript

Comments on the Quality of English Language: English grammar needs extensive improvement through proofreading.

Thank you very much for your comments. This manuscript was edited for proper English language, grammar, punctuation, spelling.

Round 2

Reviewer 2 Report

Comments and Suggestions for Authors

Sufficient improvement in the manuscript has been done by the authors. Moreover, justification of the comments raised by me has been given.

Comments on the Quality of English Language

The manuscript can be considered for acceptance after minor editing.